# Hypermutator strains of *Pseudomonas aeruginosa* reveal novel pathways of resistance to combinations of cephalosporin antibiotics and beta-lactamase inhibitors

Augusto Dulanto Chiang[1☯], Prashant P. Patil[1☯], Lidia Beka[1¤], Jung-Ho Youn[2], Adrien Launay[1], Robert A. Bonomo[3,4,5,6], Pavel P. Khil[1,2], John P. Dekker [1,2]*

**1** Bacterial Pathogenesis and Antimicrobial Resistance Unit, LCIM, NIAID, NIH, Bethesda, Maryland, United States of America, **2** Dept. Laboratory Medicine, NIH Clinical Center, NIH, Bethesda, Maryland, United States of America, **3** Department of Medicine, Case Western Reserve University School of Medicine, Cleveland, Ohio, United States of America, **4** Louis Stokes Cleveland Department of Veterans Affairs Medical Center, Cleveland, Ohio, United States of America, **5** Departments of Pharmacology, Molecular Biology and Microbiology, Biochemistry, and Proteomics and Bioinformatics, Case Western Reserve University School of Medicine, Cleveland, Ohio, United States of America, **6** CWRU-Cleveland VAMC Center for Antimicrobial Resistance and Epidemiology (Case VA CARES) Cleveland, Ohio, United States of America

☯ These authors contributed equally to this work.
¤ Current address: National Cancer Institute, NIH, Bethesda, Maryland, United States of America
* john.dekker@nih.gov

**Data Availability Statement:** The authors confirm that all data underlying the findings are fully

## Abstract

Hypermutation due to DNA mismatch repair (MMR) deficiencies can accelerate the development of antibiotic resistance in *Pseudomonas aeruginosa*. Whether hypermutators generate resistance through predominantly similar molecular mechanisms to wild-type (WT) strains is not fully understood. Here, we show that MMR-deficient *P. aeruginosa* can evolve resistance to important broad-spectrum cephalosporin/beta-lactamase inhibitor combination antibiotics through novel mechanisms not commonly observed in WT lineages. Using whole-genome sequencing (WGS) and transcriptional profiling of isolates that underwent in vitro adaptation to ceftazidime/avibactam (CZA), we characterized the detailed sequence of mutational and transcriptional changes underlying the development of resistance. Surprisingly, MMR-deficient lineages rapidly developed high-level resistance (>256 µg/mL) largely without corresponding fixed mutations or transcriptional changes in well-established resistance genes. Further investigation revealed that these isolates had paradoxically generated an early inactivating mutation in the *mexB* gene of the MexAB-OprM efflux pump, a primary mediator of CZA resistance in *P. aeruginosa*, potentially driving an evolutionary search for alternative resistance mechanisms. In addition to alterations in a number of genes not known to be associated with resistance, 2 mutations were observed in the operon encoding the RND efflux pump MexVW. These mutations resulted in a 4- to 6-fold increase in resistance to ceftazidime, CZA, cefepime, and ceftolozane-tazobactam when engineered into a WT strain, demonstrating a potentially important and previously unappreciated mechanism of resistance to these antibiotics in *P. aeruginosa*. Our results suggest that MMR-deficient isolates may rapidly evolve novel resistance mechanisms, sometimes with complex

available without restriction. All sequencing files have been deposited with NCBI under BioProject PRJNA693407. The PT sequencing files are available under BioProject PRJNA562735. Custom bioinformatic scripts have been deposited with Zenodo (DOI: 10.5281/zenodo.7223566). Requests for isolates generated in this work require a negotiated Material Transfer Agreement with the NIH and U.S. Government.

**Funding:** ADC, PPP, LB, AL, PPK, and JPD are supported by the Intramural Research Program of the National Institute of Allergy and Infectious Diseases (NIAID). RAB reports extramural funding from NIAID under Award Numbers R01AI100560, R01AI063517, and R01AI072219. RAB is also supported in part by funds and/or facilities provided by the Cleveland Department of Veterans Affairs, Award Number 1I01BX001974 from the Biomedical Laboratory Research & Development Service of the VA Office of Research and Development, and the Geriatric Research Education and Clinical Center VISN 10. The funders had no role in study design, data collection and analysis, decision to publish, or preparation of the manuscript.

**Competing interests:** I have read the journal's policy and the authors of this manuscript have the following competing interests: RAB reports grants from Entasis, Merck, Wockhardt, Shionogi, and Venatorx. None of these entities were involved in any part of this study in any way. The other authors declare no conflicts of interest.

**Abbreviations:** AMR, antimicrobial resistance; BER, base excision repair; C/T, ceftolozane-tazobactam; CZA, ceftazidime/avibactam; DEG, differentially expressed gene; MDR, multidrug resistant; MMR, mismatch repair; RND, resistance-nodulation-division; WGS, whole-genome sequencing; WT, wild type.

dynamics that reflect gene inactivation that occurs with hypermutation. The apparent ease with which hypermutators may switch to alternative resistance mechanisms for which antibiotics have not been developed may carry important clinical implications.

## Introduction

*Pseudomonas aeruginosa* is a leading cause of serious infections in humans. A distinguishing feature of this pathogen is its remarkable ability to develop resistance to most classes of antibiotics through chromosomal mutations, without the need for horizontal gene transfer [1–3]. Consequently, multidrug resistant (MDR) *P. aeruginosa* can emerge rapidly with treatment in a number of important clinical contexts, and an understanding of the mechanisms by which this resistance evolves is critical to developing the next generation of antipseudomonal agents. A frequently observed phenomenon during chronic infections with *P. aeruginosa* is the development of hypermutation, usually caused by inactivation of genes involved in DNA repair [4–6]. The resulting DNA repair deficiencies can elevate spontaneous mutations rates by 100- to 1,000-fold, with distinct mutational spectra depending on the specific DNA repair pathway affected [7,8]. Hypermutation due to deficiencies in the mismatch repair (MMR) system in particular has been shown to accelerate the emergence of antimicrobial resistance (AMR) both in vitro and in vivo and is thus of considerable clinical concern [9–11]. Hypermutator *P. aeruginosa* isolates have been found in up to 50% of cystic fibrosis respiratory specimens, and hypermutation has been linked to development of MDR phenotypes over periods of years to decades in this context [4,12–18]. Recent work has also demonstrated that hypermutation can lead to the evolution of antibiotic resistance over the time course of days in the context of acute systemic infection [19]. Though the link between hypermutation and the development of AMR is well documented, the detailed mutational steps leading to resistance to many different classes of antibiotics in hypermutators have not been well characterized [20–22].

Ceftazidime/avibactam (CZA) is an expanded spectrum antimicrobial consisting of an antipseudomonal cephalosporin (ceftazidime) and a novel beta-lactamase inhibitor (avibactam) [23–25]. The development of CZA added a valuable option against many MDR *P. aeruginosa*, given that avibactam is a potent inhibitor of the *P. aeruginosa* PDC (AmpC) cephalosporinase, which is often overexpressed in beta-lactam-resistant *P. aeruginosa* isolates [2]. However, CZA-resistant *P. aeruginosa* isolates were described within a year following its introduction, mostly associated with the development of point mutations in the chromosomal PDC cephalosporinase and overexpression of the RND-class MexAB-OprM efflux systems [26–35]. Recently, we demonstrated that MMR-deficient *P. aeruginosa* isolates with an inactivated *mutS* gene can rapidly develop high-level CZA resistance in an in vitro adaptive evolution model [19]. In that study, lineages derived from a laboratory strain *P. aeruginosa* MPAO1 (MPAO1-WT) and an MPAO1 strain containing a transposon insertion in the *mutS* gene (MPAO1-*mutS*^Tn) were passaged through a gradient of increasing CZA concentrations. While all lineages evolved CZA resistance, the MPAO1-*mutS*^Tn isolates developed clinical levels of CZA resistance more rapidly than the MPAO1-WT isolates (median passages to MIC = 16 μg/mL for MPAO1-*mutS*^Tn = 2.5, range 2 to 4; versus MPAO1-WT = 15, range 7 to 16) [19]. In the present work, we employ large-scale genomic and transcriptional analyses in combination with genetic engineering to study the mechanistic basis of resistance to cephalosporin/beta-lactamase inhibitor combination antibiotics in these MMR-deficient hypermutators.

## Results

### Wild-type isolates evolve CZA resistance through mutations in previously described gene targets

To characterize mutations that emerged under CZA selection in Khil and colleagues [19], we performed Illumina whole-genome sequencing (WGS) of evolved isolates. Three lineages (MPAO1-WT, MPAO1-*mutS*$^{Tn}$, and a comparator clinical *P. aeruginosa* isolate, PT) were grown in CZA concentrations ranging from 0.5 μg/mL to 256 μg/mL (see Methods and Khil and colleagues [19] for details). Liquid cultures that displayed detectable growth were plated, and up to 3 colonies per passage were selected for sequencing with a focus on isolates straddling the clinically relevant CZA concentrations. Following quality control, WGS data from 303 isolates were included in the analysis, with a median of 3.7 million reads and median coverage of 51× per sample (S1 and S2 Tables). To prioritize variants that were likely targets of selection and to account for potential mutations introduced during lab manipulations prior to sequencing in hypermutator strains, we focused on SNVs that were retained in terminal isolates as CZA resistance evolved. These "fixed" variants were defined as those that were found in at least 2 isolates of a given lineage including at least 1 terminal isolate, but that were not present in the parental (starting) strain of the lineage. For this purpose, the set of variants present in the parental MPAO1-*mutS*$^{Tn}$ was assessed by sequencing of the bulk starting bacterial stock solution and 6 isolates spanning 2 of the lineages at the end of the first passage day. This approach was chosen to take into account the high spontaneous mutation rate in these lineages and revealed 54 "early" variants that were dominant before the end of the first passage day (S1 Data, Methods). Notably 50/50 (100%) of the early SNVs that occurred were transitions, consistent with a MutS deficiency-driven hypermutation spectrum [36–39].

We identified a total of 14, 21, and 139 fixed variants that emerged under CZA selection in the PT, MPAO1-WT, and MPAO1-*mutS*$^{Tn}$ lineages, respectively, representing a mean of 3.5, 5.3, and 46.3 fixed variants per lineage (Figs 1A and S1). In agreement with previous work [36–39], fixed mutations in the MutS-deficient lineages demonstrated a strong transition bias (119/120 SNV), compared to the WT lineages (5/11 SNV; Fisher's exact test *p*-value < 0.0001). To evaluate the global properties of the observed fixed mutations, we applied a comparative genomics approach using the *Pseudomonas* ortholog database and compared average amino acid identity versus gene conservation for the coding sequences with non-synonymous substitutions [40,41] (see Methods). This analysis demonstrated a broad distribution of mutated genes in the hypermutator lineages and confirmed that mutations were not confined to poorly conserved accessory genes but were distributed across genes at all conservation levels (Fig 1B). Further Gene Ontology analysis of the variants in the hypermutator lineages demonstrated enrichment in a number of functional classes including purine nucleotide metabolism and cell wall biosynthesis (S2 Fig).

The majority of fixed variants in lineages from both WT strains (16/21 in MPAO1-WT and 9/14 in PT) were located within genes previously described in association with beta-lactam resistance, including efflux pumps and their transcriptional regulators (*mexR*, *MexA*, *MexB*, *nalD*, *mexE*, *mexF*, *oprN*) or in the Ω-loop of the PDC chromosomal cephalosporinase (Fig 2 and S3 Table). Of note, a 7 kilobase deletion involving 8 genes, including part of *mexF*, *oprN*, PA2496, PA2497, PA2498 (*yahD*), PA2499 (*ykoA*), PA2500 (*cynX*), PA2501, and part of PA2502 (PAO1 Reference coordinates 2,812,525 to 2,819,849), emerged in passage 6 in the MPAO1-WT 1D lineage. Lineage 3B in the PT isolate acquired a 21-nucleotide deletion in the gene encoding the PDC beta-lactamase gene, which has previously been demonstrated to confer CZA and ceftolozane-tazobactam (C/T) resistance [27]. Mutations in the ATP-binding subunit of the Clp protease *clpA*, as well as in the PhoP/PhoQ phosphorelay system were also

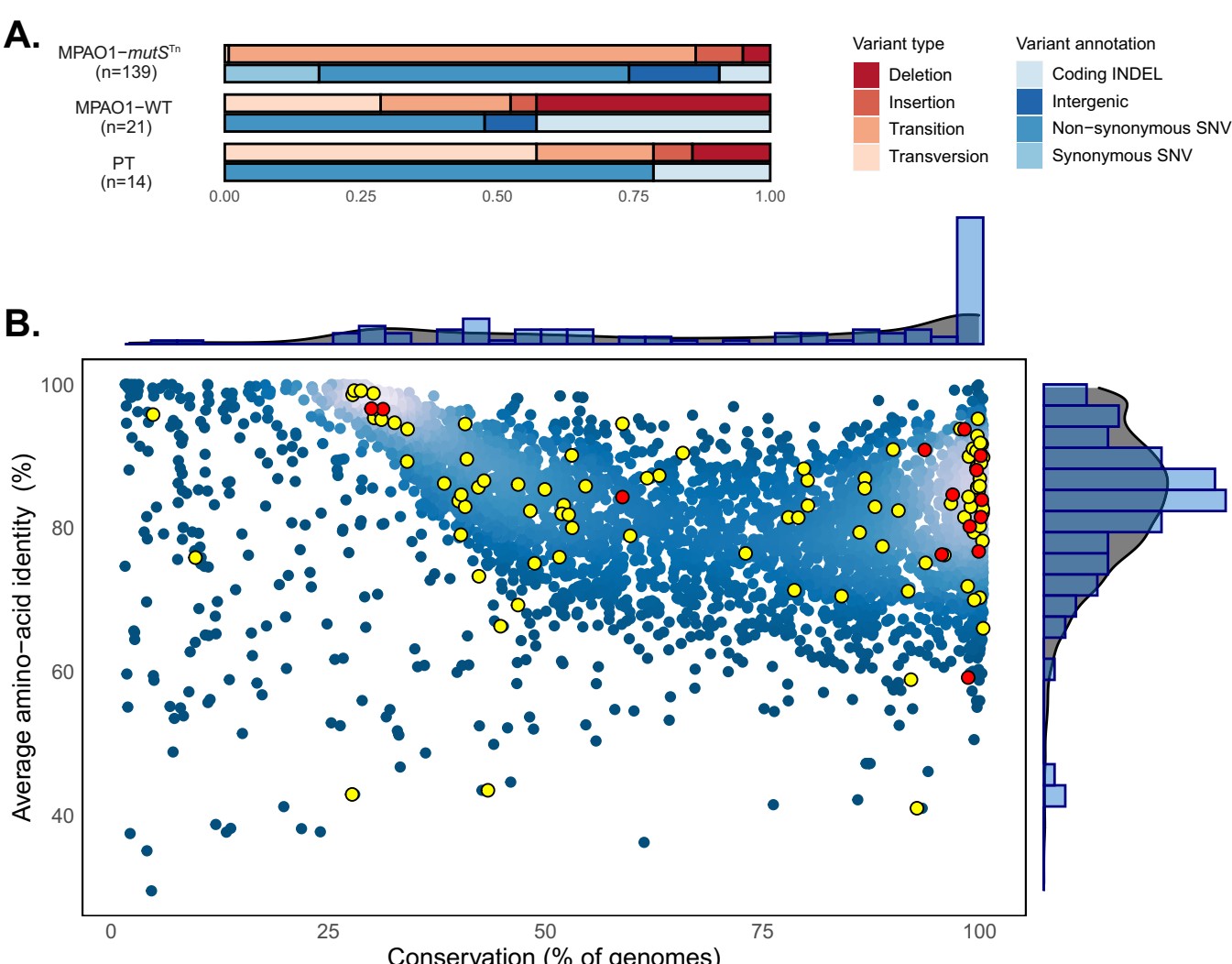

**Fig 1. General characteristics of fixed genomic variants that emerged during CZA selection.** (A) Proportions and types of fixed variants acquired during the course of CZA adaptation for each genotype. The total number of fixed variants acquired across lineages per genotype is indicated in parenthesis. (B) Evolutionary conservation of genes with fixed variants acquired in MPAO1-*mutS*$^{Tn-}$ (yellow) and MPAO1-WT (red) CZA adaptation experiments in the *Pseudomonas* genus. Each *P. aeruginosa* PAO1 CDS is plotted as a dot representing average amino acid identity and the proportion of genomes containing the given CDS (breadth of coverage) among a set of 100 complete *Pseudomonas* genomes (see S6 Data and Methods). Genes without fixed mutation are represented in blue. The marginal density plots and histograms show the distribution of a corresponding conservation measure in all CDSs (gray) and genes containing fixed variants (blue), respectively. The underlying data to generate this figure can be found in S2 Data. CZA, ceftazidime/avibactam; WT, wild type.

observed in independent lineages (*clpA* in all WT, PT, and *mutS* lineages except WT lineage 1B; *phoQ* in lineages 1A, 1D, 2A, 2B, and 2D, see Fig 2 and S3 Table).

## MMR-deficient isolates evolved CZA resistance through mutations in genes outside of the established cephalosporin resistome

We next looked at the fixed variants in the MPAO1-*mutS*$^{Tn}$ lineages as CZA resistance evolved. Strikingly—and in contrast to the behavior of the WT isolates—almost all fixed variants acquired as CZA resistance evolved through the clinical breakpoint of 16/4 µg/mL were located in genes that have not been previously described to be involved in CZA or third generation cephalosporin resistance (Fig 2). Though mutations in these lineages did not occur in the

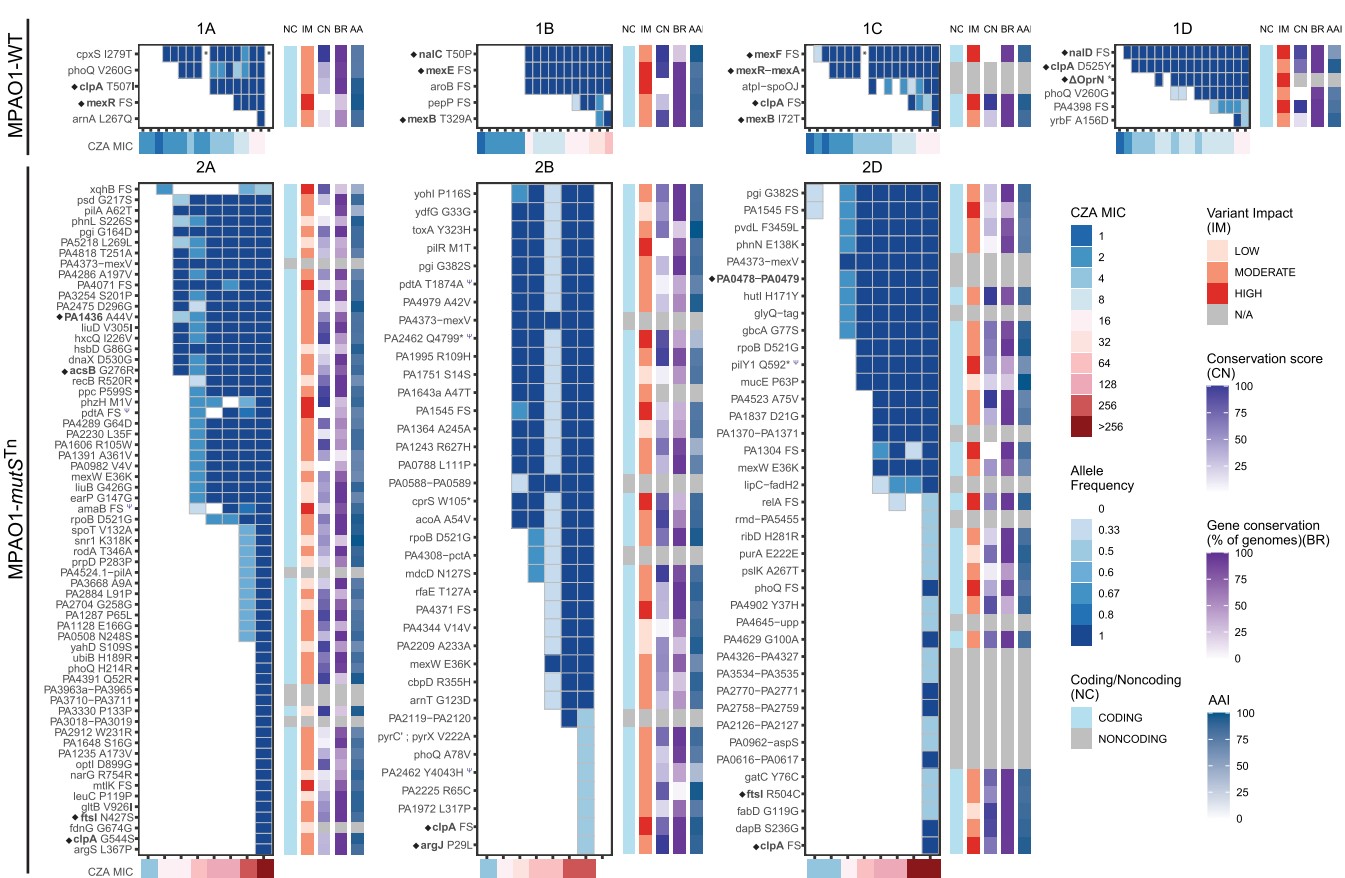

**Fig 2. Wild-type lineages demonstrated mutations in canonical resistance genes, while MMR-deficient lineages developed mutations in genes not previously associated with cephalosporin resistance.** Fixed variants are plotted versus passage (horizontal axis tick marks) and CZA MIC (µg/mL), with a filled tile indicating presence of the variant at the corresponding passage. The shade of blue represents the proportion of isolates (allele frequency) in a given passage carrying the corresponding variant. Up to 3 isolates per lineage per passage underwent WGS. The sidebars to the right of each plot represent functional characteristics of the variant and its protein targets as follows: coding/noncoding (NC) variant; variant impact (IM) as predicted by SnpEff; conservation score (CN) as relative conservation of a mutated AA position within each protein defined as a within-CDS percentile of the Jensen–Shannon divergence scores calculated over a set of 100 *Pseudomonas* genomes; breadth (BR) as the percentage of 100 *Pseudomonas* genomes in which the CDS is present; average AAI of the CDS with respect to its orthologs. Genes marked with a Ψ symbol were also mutated in the no antibiotic control experiments. Genes marked with a diamond ◆ are those for which experimental evidence supports a role in third generation cephalosporin resistance. An asterisk (*) represents a passage with no sequenced isolates. FS = Frameshift variant. ΔOprN = 7-kb deletion including the *oprN* gene (partial deletion of *mexF*, *oprN*, PA2496, PA2497, PA2498 (*yahD*), PA2499 (*ykoA*), PA2500 (*cynX*), PA2501 and part of PA2502; PAO1 Reference coordinates 2,812,525–2,819,349). CZA MIC was determined by E-test. The underlying data to generate this figure can be found in S2 Data. AAI, amino acid identity; CZA, ceftazidime/avibactam; MMR, mismatch repair; WGS, whole-genome sequencing.

"classical" resistance genes, we identified 9 mutations across 6 genes for which some level of prior experimental evidence exists supporting a role in resistance to third generation cephalosporins: PA1436 (*mexN* of MexMN efflux pump), *acsB*, *ftsL*, *clpA*, *argJ*, PA0478-PA0479 [30,42–47]. All of these mutations first appeared in passages with CZA MIC equal to or greater than the clinical resistance breakpoint of 16 µg/mL, and thus contributed to resistance only at these higher MICs. Notably, one of the 54 "early" variants present in the parental strain of the MPAO1-*mutS*[Tn] lineages was located within MexB (W753R). This variant, discussed more fully below, was present by the end of the first day of passage in all sequenced isolates.

Given the high mutation rates in the hypermutators, a large number of background mutations that do not improve fitness mutations are expected to be co-selected along with mutations that improve fitness. We thus expect that some proportion of the observed fixed variants (possibly the majority) did not contribute directly to antibiotic resistance. As a first step

towards understanding which subset of fixed variants in the MPAO1-*mutS*$^{Tn}$ isolates were those contributing to antibiotic resistance, we extended the comparative genomic analysis by adding measures of relative conservation and predicted impact at each position [40,48] (Fig 2, Methods). This analysis predicted a moderate to high impact for a number of fixed variants in the MPAO1-*mutS*$^{Tn}$ lineages, indicating a number of potential functional targets. An additional indicator of the importance of a specific variant or gene target for CZA resistance is independent selection and fixation in different lineages. Overall, 8 genes were mutated in more than 1 MPAO1-*mutS*$^{Tn}$ lineage: PA1545 (hypothetical protein belonging to *Pseudomonas* ortholog group 3815) [40]; glucose-6-phosphate isomerase, *pgi*; *pdtA* (phosphate depletion regulated TPS partner A); the β-subunit of RNA polymerase, *rpoB*; *mexW* (subunit of the MexVW RND-type efflux pump); the *phoQ* subunit of the 2-component sensor PhoP/Q; *ftsI* (PBP3); and the ATP-binding subunit of the Clp protease, *clpA* (S3 Fig). An additional noncoding variant was shared across the 3 lineages, located 82 nucleotides upstream from the MexVW operon (position 4903384 in the PAO1 reference). Five of the variants in these 8 common targets (RpoB D521G, Pgi G164D, a frameshift in PA1545, MexW E36K, and a SNV 82nt upstream from MexV) were identical between at least 2 lineages, suggesting that these variants may have been present at low proportions in the heterogeneous starting population from which each lineage was derived in the microevolution experiment, as opposed to independent de novo occurrence during adaptation. Another possibility is early-stage cross-contamination between lineages in the experiment. We were unable to find supportive genomic evidence identifying a cross-contamination event to explain the sharing of these mutations, though this case may be hard to distinguish from a variant that is shared by the starter populations at low frequency. Nevertheless, the fact that these variants had initially low population frequency but subsequently became dominant in multiple lineages under increasing CZA selection suggests that they may have provided fitness advantages.

To examine whether some of the observed variants may have been selected by conditions in the serial passaging experiment other than antibiotic pressure, we compared variants that occurred under CZA selection to the variants that occurred under the same conditions in the absence of CZA for 11 sequential passages. We found 5 shared genes with at least 1 variant among isolates passaged with and without CZA (S4 Table). Out of 12 variants in these 5 shared genes (*pdtA*, *amaB*, PA2462, *pilY*, and *rfaE*), only 1 variant was identical between isolates passaged with and without CZA (G insertion at position 1114264, in the *amaB* gene). The *pdtA* and PA2462 genes have unusually long open reading frames (>10 kb), which increases the likelihood of observing repeated mutations in these genes by chance and might explain why mutations were observed in these genes under both conditions. The 5 variants seen in *pdtA*, *amaB*, PA2462, and *pilY* consisted of short indels in GC-rich contexts, which undergo higher rates of mutation in the mismatch-repair deficient *P. aeruginosa* isolates [49,50]. None of these variants were shared between multiple evolved CZA-resistant lineages.

To explore whether the CZA-adapted MMR-deficient isolates demonstrated gross growth defects due to mutational accumulation, we measured growth curves in LB broth for CZA-evolved isolates at different points along antibiotic adaptation from all 3 MPAO1-*mutS*$^{Tn}$ lineages and 1 MPAO1-WT lineage for comparison (Methods, S4 Fig). While there is a reduction in the growth rate in lineage 2D at passage 12 and a progressive trend to slower growth along passages in lineage 2A, the growth curves of most isolates were not substantially different from those of MPAO1-WT 1A.

## Transcriptional profiling suggests candidate novel mechanisms conferring resistance in MMR-deficient hypermutator strains

Given the surprising finding that the CZA-resistant MPAO1-*mutS*$^{Tn}$ lineages largely lacked fixed mutations in genes known to be involved in CZA and third generation cephalosporin

resistance, we next sought to examine whether the observed mutations had resulted in overexpression of known resistance genes such as the MexAB-OprM efflux pump and PDC beta lactamase through unexpected transcriptional control mechanisms. We thus performed RNA-seq on 3 evolved CZA-resistant MPAO1-WT isolates representing 3 different WT lineages, as well as on a series of MPAO1-*mutS*$^{Tn}$ isolates with different MICs over the clinically relevant range from the 3 hypermutator lineages and their respective ancestors. At total of 61 mid-log phase RNA-seq datasets were obtained from 22 isolates with an average of 6.1 million reads per transcriptome (S3 Data).

Comparison of transcriptomes demonstrated substantial differences between the evolved CZA-resistant MPAO1-WT and MPAO1-*mutS*$^{Tn}$ isolates. While the overall numbers of differentially expressed genes (DEGs) were not significantly different between the terminal WT and MMR-deficient groups (S5 Fig), principal component analysis (S6 Fig) demonstrated that their content differed, and transcriptomes from the 2 groups of isolates were well separated in the first 2 principal components. These global differences in DEG content between WT and MMR-deficient lineages is also evident from a heatmap representation (Fig 3A). Analysis of expression of known resistance genes revealed up-regulation of the MexAB-OprM efflux pump components in the MPAO-WT transcriptomes (*mexA* >1.5 and *mexB* >1.3 log$_2$ fold change (LFC) in all 3 WT lineages versus WT ancestor, $p < 0.001$) (Fig 3B). These expression changes provide an explanation for CZA resistance in the WT isolates that is consistent with both initial genomic analysis and previously described mechanisms. In contrast, the MPAO1-*mutS*$^{Tn}$ isolates did not show changes in expression of *mexA*, *mexB*, *oprM* or in other previously described major resistance genes, including *oprD*, *ftsI*, or the PDC beta-lactamase gene (Fig 3B). Additionally, there were no baseline differences in expression of the 6 major resistance genes listed above between the MPAO1-WT and MPAO1-*mutS*$^{Tn}$, using thresholds of p$_{adjust}$ < 0.01 and/or LFC >1 (S7 Fig). To narrow down possible mechanisms conferring CZA resistance, we identified 91 genes that were consistently up-regulated and 142 consistently down-regulated in MPAO1-*mutS*$^{Tn}$ lineages (S8 Fig). This analysis revealed an important detail: The MexVW operon demonstrated expression with early evolution of CZA resistance in all 3 MPAO1-*mutS*$^{Tn}$ lineages, and this coincided with acquisition of fixed mutations in this operon, including the E36K mutation in MexW with moderate to high conservation and impact scores (Fig 2). For comparison, we examined the expression of other RND transporter systems in these isolates and found that MexVW was the only transporter that was overexpressed consistently across lineages (Fig 3C). This increased expression of *mexV* and *mexW* was statistically significant when comparing the MPAO1-*mutS*$^{Tn}$ terminal isolates with the MPAO1-WT terminal isolates (LFC > 3; p$_{adjust}$ < 4 × 10$^{-15}$). We also noted that *mexG*, *mexH*, and *mexI* were down-regulated in all MPAO1-*mutS*$^{Tn}$ lineages.

## Mutations in the MexVW efflux pump operon confer 4- to 6-fold increase in broad spectrum cephalosporin resistance in an isogenic background

Based on the foregoing combination of genomic and transcriptomic analysis, we focused on the MexVW efflux pump as potentially contributing to CZA resistance in the hypermutators. To test the effects of the 2 identified mutations, we introduced them individually and in combination into an MPAO1-WT strain to generate 3 derivative strains: MPAO1 *mexV* -82:T>C, MPAO1 MexW E36K, and MPAO1 *mexV* -82:T>C + MexW E36K. We performed susceptibility testing to a variety of classes of antimicrobials and categories of beta lactams on all 3 strains (Figs 4A and S9 and S5 Table). This demonstrated that each mutation individually conferred a 1.5- to 2-fold increase in CZA MIC, and the 2 mutations in combination conferred a statistically significant 4- to 6-fold increase in MICs to ceftazidime, CZA, C/T, and cefepime.

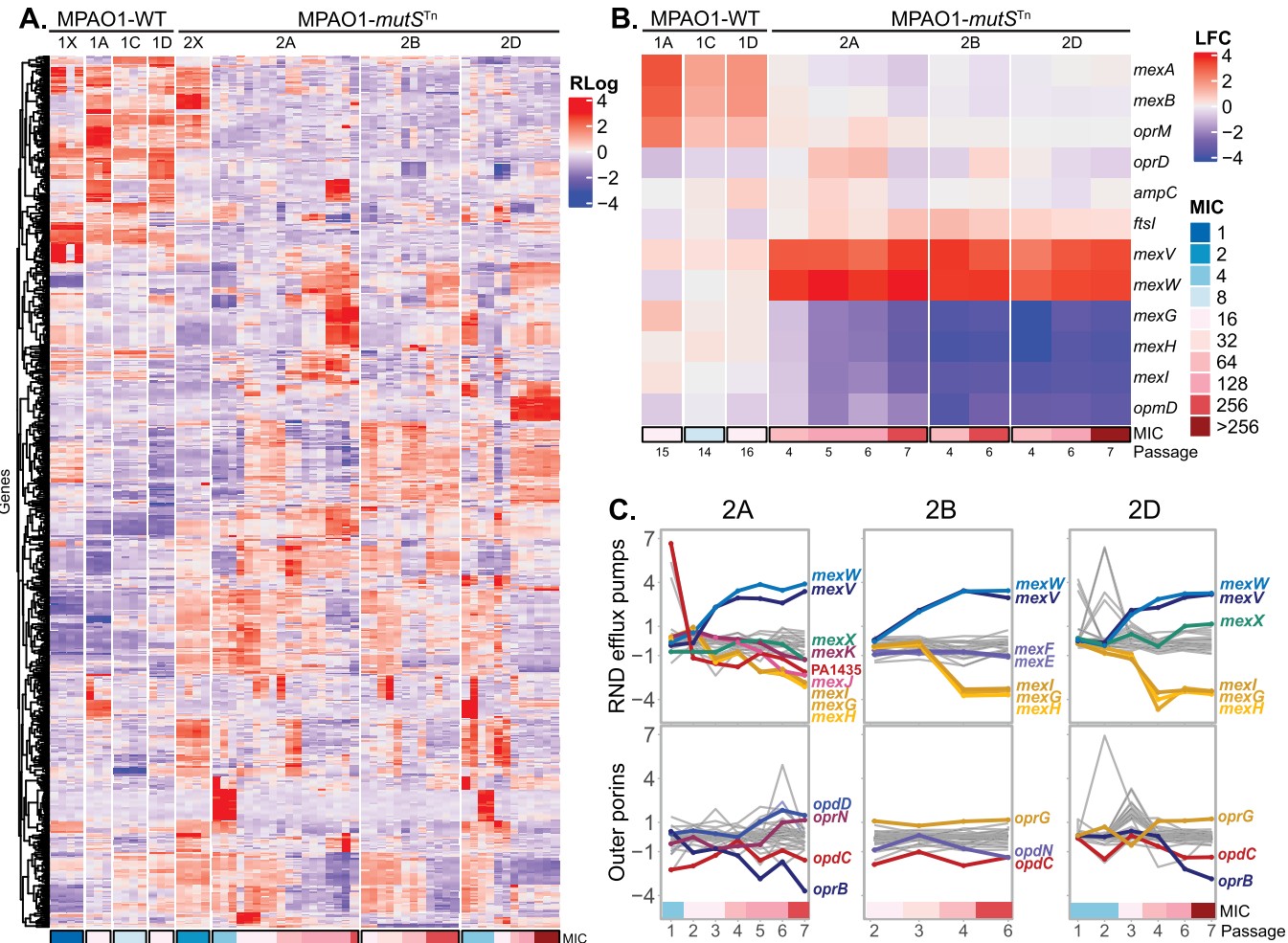

**Fig 3. Dissimilar transcriptome profiles highlight lack of involvement of major cephalosporin resistance genes in CZA resistance in MMR-deficient lineages and demonstrate MexVW overexpression.** Three lineages of MPAO1-WT and MPAO1-*mutS^Tn* were selected for transcriptomic analysis and 3 biological replicates from 1 isolate per passage underwent RNA sequencing. (A) A clustered heatmap of the top 750 most variable genes across CZA-resistant isolates is shown. Rlog normalized levels of expression of each gene (rows) were scaled within each row. Isolates (columns) are grouped by lineage and arranged by genotype with WT on the left and hypermutators on the right. Within each lineage, isolates are arranged from left to right in order of increasing number of passages under CZA selection, with the corresponding MIC indicated at the bottom. 1X and 2X represent the MPAO1-WT and MPAO1-*mutS^Tn* parental strains of the respective lineages. (B) LFC with respect to their corresponding ancestor of the terminal MPAO1-WT isolates (first 3 columns) and CZA-adapted MPAO1-*mutS^Tn* isolates (last 9 columns). The rows represent 6 major beta-lactam resistance genes in *P. aeruginosa* from 4 major resistance pathways (encoding the MexAB-OprM efflux pump, OprD outer membrane porin, PDC beta-lactamase, and penicillin-binding-protein 3 FtsI). Data for 2 differentially expressed RND efflux pumps in the MPAO1-*mutS^Tn* lineages (MexVW and MexGHI-OpmD) are additionally shown. (C) Expression levels of RND efflux pumps and outer membrane porins in MPAO1-*mutS^Tn* lineages during the course of the CZA adaptation experiment. The vertical axis represents the LFC of expression levels compared to the early isolates, and the horizontal axis shows the passage number. CZA MIC is indicated with color tiles under the axis. Lines are colored and labeled if the respective gene reached a LFC > 1.0 in the terminal isolates of the lineage. The underlying data to generate this figure can be found in S2 Data. CZA, ceftazidime/avibactam; MMR, mismatch repair; WT, wild type.

Additional testing demonstrated no changes in susceptibility to aminoglycosides, meropenem or fluoroquinolones with respect to WT MPAO1 (S5 Table and S9 Fig). Standard tests of AMR were supplemented with an assessment of growth kinetics in the presence of CZA. These results corroborated the ability of engineered mutants to grow under higher CZA concentrations, with the double mutant being able to grow at CZA concentrations 4 times that of the wild type (Fig 4B). The relative contributions of the *mexV* -82:T>C and MexW E36K to resistance to each of the tested antibiotics are displayed in Fig 4C. Extended antimicrobial

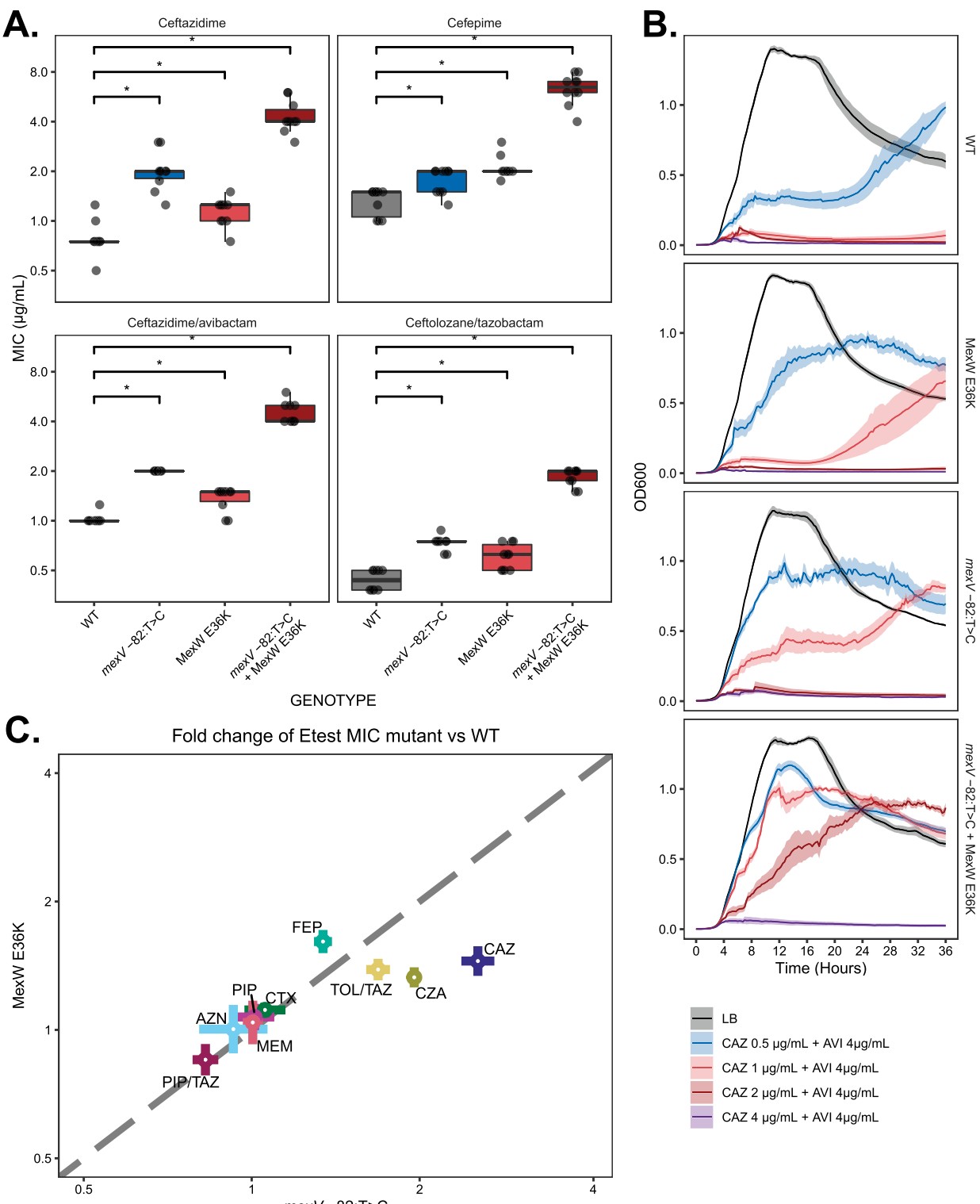

**Fig 4. MexVW efflux pump mutations introduced into a WT MPAO1 strain confer resistance to cephalosporins and cephalosporin/beta-lactamase combination antibiotics.** (**A**) Gradient diffusion (E-test) MICs for MPAO1-WT, *mexV* -82:T>C, MexW E36K, and double mutant (*n* = 10 per antibiotic/genotype combination). The boxes display the median with the lower and upper hinges corresponding to the first and third quartiles. Brackets with asterisks above the plot indicate statistical significance with respect to MPAO1-WT (Wilcoxon 2-sided *p*-value < 0.05). All pairwise comparisons between mutants were statistically significant as well (S7 Table). (**B**) Growth curves of MexVW engineered strains at

37˚C in LB broth with a range of ceftazidime concentrations with fixed 4 μg/mL avibactam concentration. Lines represent the median $OD_{600}$ of 7 biological replicates per time point, with shaded envelope representing the standard error of the mean. (**C**) Scatterplot showing the relative contribution of the *mexV* -82:T>C and MexW E36K mutations to resistance to each antibiotic. Each colored point represents the mean ratio of gradient diffusion MIC in the corresponding mutant strain to that in the parental MPAO1 strain, with error bars represent the corresponding standard error of the mean. A dashed line of equality is included. The underlying data to generate this figure can be found in S2 Data. AZN, aztreonam; CAZ, ceftazidime; CTX, cefotaxime; CZA, ceftazidime/avibactam; FEP, cefepime; MEM, meropenem; PIP, piperacillin; PIP/TAZ, piperacillin/tazobactam; C/T, ceftolozane/tazobactam; WT, wild type.

susceptibility testing was also performed on the evolved hypermutator isolates for comparison, with the caveat that interpretation is complicated by the growth of satellite colonies within the zones of inhibition (see S10 Fig and S6 Table).

## Rarity of the observed *mexV* and *mexW* mutations in public genome data suggests many pathways to resistance are available to hypermutators

Given that mutations in the genes encoding the MexVW efflux pump have not commonly been associated with resistance to cephalosporin/beta-lactamase inhibitor combinations, we sought to understand the frequency with which these variants occurred in publicly available *P. aeruginosa* genomes. BLAST was used to search for sequence homologues of the *mexW* CDS and intergenic sequence upstream from *mexV* in a set of 7,492 *P. aeruginosa* genomes from the NCBI Pathogen Detection database [51] (see S8 Table and Methods). Neither of the 2 specific variants identified in our study was found in this search, and the neighboring amino acid and nucleotide sequences were relatively conserved (S8 Table). The apparent rarity of the specific mutations uncovered in this adaptive evolution experiment suggests that sequence space may harbor a large universe of potential resistance mechanisms available to hypermutators.

## Early inactivating MexB W753R mutation may have driven an evolutionary search for alternative resistance mechanisms

As noted above, the MPAO1-*mutS*$^{Tn}$ strain was found to differ from the MPAO1-WT strain by 54 variants (50 SNVs and 4 indels), with 33 of these variants present at a majority frequency in the starting stock and an additional 21 that occurred during the first passage, but before CZA resistance appeared (S1 Data). All 50 of the SNVs were transitions, consistent with the expected outcome of MutS-deficient hypermutation. Though these mutations in aggregate do not account for the CZA resistance that subsequently developed, it was possible that they may have played unappreciated roles that influenced the likelihood of evolving resistance through major efflux pathways. Importantly, these mutations included a transition variant in the *mexB* gene in the MPAO1-*mutS*$^{Tn}$ isolate (resulting in the W753R substitution), and we hypothesized that this substitution may have inactivated the MexAB-OprM efflux pump, blocking further evolution that would have used up-regulation of this transporter to confer resistance. To test whether this mutation would have been compatible with the MexAB-OprM-based CZA resistance that ultimately evolved in the WT strains, we introduced it into 3 different strains: (1) an MPAO1-WT strain; (2) an evolved CZA-resistant WT isolate with increased MexAB-OprM expression (1C12-1); and (3) an engineered MPAO1 *mexV* -82:T>C + MexW E36K background. The aforementioned 1C12-1 isolate was obtained during serial CZA passage and contains an intergenic SNV between *mexR* and *mexA* (471932 C>T), upstream from the MexAB-OprM operon. This intergenic variant appears to be responsible for the increase in MexAB-OprM expression seen in this lineage by RNA-seq (Fig 3B).

The introduction of the MexB W753R substitution significantly reduced the aztreonam MIC in all 3 strains, consistent with disruption of MexAB-OprM function. In the

MexAB-OprM-overexpressing 1C12-1 isolate, the introduction of MexAB W753R also decreased the CZA and ceftazidime MIC to pre-evolved WT baseline levels (S11 Fig), suggesting that this isolate's CZA and CAZ resistance were entirely due to MexAB-OprM overexpression, and that the MexB W753R mutation was fully inactivating. These findings support the hypothesis that MexB W753R was not compatible with the evolutionary mechanisms the WT isolates used to generate resistance to CZA under selection, and suggests that this early inactivating mutation in the MexAB-OprM efflux pump, a primary mediator of CZA resistance, potentially drove an evolutionary search for alternative resistance mechanisms. Interestingly, the CZA and CAZ resistance were not affected by this *mexB* mutation when introduced in a *mexV* -82:T>C + MexW E36K background (S11 Fig), providing further support that MexVW can confer cephalosporine resistance through a MexAB-OprM-independent mechanism.

## *P. aeruginosa* hypermutators and MexAB-OprM-deficient assemblies are common in public genome databases

To evaluate how commonly hypermutation phenotypes and MexAB-OprM efflux pump deficiency occur in *P. aeruginosa*, we used the NCBI Pathogen Genome Database to search for disruptive mutations (e.g., frameshifts and stop codons, mutations that result in complete loss or severe truncation of the affected protein) in the corresponding genes (see Methods). Inactivation of a number of different genes can result in hypermutator phenotypes in *P. aeruginosa*, including MMR genes *mutS*, *mutL*, and *uvrD*, base excision repair (BER) genes *mutT*, *mutM* and *mutY*, and others. We identified severely disruptive variants in at least one of these genes in 260 out of 6,805 assemblies (3.8%) (S9 Table). Given that some isolates can be clonal, and this might inflate this estimate, a conservative lower bound estimate of 155/6,805 assemblies (2.3%) was calculated assuming that all repeated variants are clonal. We similarly estimated the frequency of disruptive variants in *mexB*, *mexA*, or *oprM* that would be expected to result in truncated or mistranslated proteins with functional inactivation of the MexAB-OprM efflux pump. We found 86 unique disruptive variants in the *mexA* gene, 148 in the *mexB* gene, and 33 in the *oprM* gene (S10 Table). In total, 629 (9.2%) isolates had a variant predicted to be severely disruptive in at least 1 of the 3 genes resulting in a nonfunctional MexAB-OprM complex. Again, a conservative lower bound estimate of 267/6,820 (3.9%) assemblies was calculated assuming that all isolates carrying a given variant represented a single clonal lineage.

A caveat to these calculations is that they are based only on NCBI genome sequences without functional verification, and we cannot rule out sequencing errors and assembly artifacts, though the prevalence would have to be extraordinarily high to explain these findings. Another caveat is that isolates from cystic fibrosis respiratory specimens, where hypermutators are common, may be overrepresented in the genomic databases [4,12–18]. However, even if hypermutators and MexAB-OprM efflux deficient isolates are somewhat less common than our estimates of ≥2.3% and ≥3.9%, this would still suggest that the co-occurrence of hypermutation and preexisting MexAB-OprM functional deficiencies may be substantially more common than appreciated. We thus asked how many assemblies in the database would be predicted to be both putative hypermutators and have a nonfunctional MexAB-OprM. Surprisingly, 91 out of 260 putative hypermutators (35%) were also found to have at least 1 sequence variant suggestive of an inactive MexAB-OprM, raising the possibility that hypermutation and MexAB-OprM inactivation may in fact occur frequently together. Once again, in order to adjust for clonality in this subset of assemblies, we conservatively assumed each individual disruptive variant ($n = 47$) to represent a single clonal lineage, obtaining a lower bound estimate of 47/260 (18%). This analysis suggests that concurrent hypermutator phenotypes and MexAB-OprM efflux pump disruption coexist in a small, but easily detectable, proportion

of the clinically relevant *P. aeruginosa* population. These isolates may be expected to evolve alternative resistance mechanisms under selection as did the isolates in the experiments in this work. However, detailed examination of resistance mechanisms in these isolates is limited due to lack of susceptibility testing data. Out of 7,492 isolates in the NCBI Pathogen Detection database, only a small minority (149 genomes, 2%) had any antimicrobial susceptibility data, with 27 (0.3%) having CZA susceptibility data available. Only 7 out of the 260 putative hyper-mutator isolates as defined above had AST phenotypic data available, and only 1/7 had CZA data. These data do not allow specific conclusions about resistance phenotypes in these isolates to be drawn.

## Discussion

In this work, we have employed in vitro adaptive evolution in combination with genomic sequencing, transcriptional profiling, and genetic engineering to study how resistance to ceph-alosporin/beta-lactamase inhibitor combinations evolves in MMR-deficient *P. aeruginosa*. Sequencing of in vitro evolved CZA-resistant MMR-deficient isolates revealed that—in con-trast to WT isolates—almost all fixed variants were located within genes not previously associ-ated with cephalosporin resistance. Additionally, while WT isolates demonstrated significant up-regulation of the MexAB-OprM efflux pump, providing an explanation for CZA resistance [52,53], the MMR-deficient isolates did not show altered expression of MexAB-OprM or of 6 other major beta-lactam resistance genes. Further investigation revealed that these isolates had paradoxically generated an early inactivating mutation in the MexAB-OprM efflux pump, likely driving an evolutionary search for alternative resistance mechanisms. The mutation was a transition variant within the mutational spectrum expected for *mutS* deficiency, similar to the 49 other early SNVs, and therefore likely directly acquired as a consequence of the MutS-deficient hypermutation. The affected residue in MexB is located in the second large periplas-mic loop of the protein, in a region postulated to be important for trimerization [54], poten-tially explaining its effect on function.

In addition to a number of other changes, expression of the less well-studied MexVW efflux pump machinery was determined to be up-regulated, and mutations were present upstream of the MexVW operon and within the coding sequence of MexW. Engineering of these 2 muta-tions in a WT genetic background increased the MICs to ceftazidime, CZA, C/T, and cefepime by 4- to 6-fold, demonstrating a previously unappreciated mechanism of resistance to cephalo-sporin/beta-lactamase inhibitor combination antibiotics in *P. aeruginosa*. These observations suggest that the MexVW efflux pump operon can be mutationally modified and used as a com-pensatory resistance mechanism to overcome MexAB-OprM loss or inhibition. Recent studies have also shown that *P. aeruginosa* strains lacking 6 major RND efflux pumps *ΔmexAB-oprM*, *ΔmexCD-oprJ*, *ΔmexX*, *ΔmexEF-OprN*, *ΔmexJKL ΔtriABC* can utilize the minor RND efflux pumps including MuxABC-OpmB and MexGHI-OpmD to gain resistance to certain antibiot-ics [55], and other recent work has demonstrated the previously unappreciated role of several less studied efflux pump complexes in conferring resistance to novel beta-lactam/beta-lacta-mase inhibitor combinations [56]. In aggregate, these findings may have implications for the development of efflux pump inhibitors for clinical use.

The MexVW complex is a member of the resistance-nodulation-division (RND) family of efflux pumps and was originally described in association with resistance to fluoroquinolones, tetracyclines, chloramphenicol, cefpirome, and erythromycin by Li and colleagues [57], but has not been previously described as involved in conferring resistance to cephalosporin/beta-lactamase inhibitor combinations such as CZA or C/T. The *mexV* and *mexW* genes appear to form a single operon without an adjacent outer membrane porin gene, though the work of Li

and colleagues suggested an association with OprM to form a tripartite MexVW-OprM complex [57]. Our analysis of transcription uncovered a progressive increase in expression of the components of the MexVW efflux pump with increasing MIC that was not observed in the WT counterparts. The exact mechanism by which the MexVW mutations examined in this study confer resistance to cephalosporins remains to be defined, but direct efflux by this pump appears likely. In addition to the up-regulation of MexVW, we observed down-regulation of the MexGHI-OpmD RND efflux pump across the MMR-deficient lineages. This pump is under the control of the redox-responsive SoxR regulon [52], but a potential role in CZA resistance would require further evaluation.

Comparison of the other mutations observed in our study with previous publications provides additional insights. Work on in vitro evolution of ceftazidime (CAZ) and CZA resistance in *P. aeruginosa* has been performed in a PA14 genetic background by Sanz-Garcia and colleagues [30]. While strains in their work developed many variants in previously described resistance genes (including *nalD*, *mexB*, *ftsI*, and large chromosomal deletions containing the MexXY-OprM operon [1,2,33,58–61]), the authors describe genes that were targeted only in lineages exposed to CZA but not CAZ. These included the RND efflux pump components *mexM* (PA14_45910) and *mexN* (PA14_45890). We observed a relatively conservative A44V substitution within the *mexN* (PA1436) gene in one of the hypermutator lineages. Determining whether the MexMN efflux pump plays a role in CZA resistance would require further study.

Other work in WT and MMR-deficient backgrounds also reported an enhanced rate of evolution of resistance. A study by Cabot and colleagues [8] analyzed variants that developed following in vitro exposure to CAZ (without avibactam), meropenem, and ciprofloxacin. Ceftazidime-resistant isolates acquired mutations in targets previously described in association with beta-lactam resistance, including *dacB*, *ampD*, PDC, *ampR*, and *galU* [8]. These ceftazidime-resistant isolates largely did not acquire cross-resistance to carbapenems, consistent with an increase in the PDC beta-lactamase expression as a mechanism of resistance. Notably, this evolutionary pathway to resistance may be less available in the presence of a beta-lactamase inhibitor such as avibactam. There were no exact variants shared between the CAZ-resistant isolates observed in our study and those of Cabot and colleagues, though we did observe mutations in shared genes: *dnaX*, *fdnG*, *pgi*, and *pvdL*. As these genes were mutated in the background of a very large number of total mutations, their role in CAZ resistance would need to be further studied before any conclusions are drawn regarding their potential contribution. Similarly, in a study by Gomis and colleagues [62], isolates passaged under selection with imipenem and imipenem-relebactam evolved mutations in previously described genes associated with beta-lactam resistance, in particular nonsense mutations in *oprD*. The addition of relebactam decreased the rate of acquisition of resistance and was associated with an increase in *mexB* expression as opposed to an increase in PDC expression as observed in the imipenem-resistant isolates. This finding would again be consistent with the effect of the beta-lactamase inhibitor on the PDC beta-lactamase, impeding opportunities for the development of resistance through this pathway, similar to conditions seen in our experiments. Indeed, in the presence of PDC inhibition, we observed evidence that MexAB-OprM overexpression contributed substantially to resistance in the WT isolates, a pathway to resistance not available to the hypermutators with the MexB W753R substitution.

In this study, we performed susceptibility testing with CZA on 1 sequenced isolate per passage to establish MICs by a standard method (E-test). Discrepancies were present between the MIC calculated from the highest CZA concentration in which growth of the population met the $OD_{600}$ threshold in the serial passage experiment and the MICs of individual isolates cultured from this passage (S4 Data). In these cases, the isolate MIC (measured by E-test) was

usually higher than the MIC calculated from the greatest CZA concentration in which there was growth passing the threshold, most often by a 2-fold dilution, though in some cases more substantial discordances were observed. These discrepancies are in part expected, as the serial passage experiment used different conditions than those used for the E-test (growth in LB broth with a quantitatively different starter inoculum, measured with an arbitrary $OD_{600}$ threshold cutoff). Additionally, E-tests were performed on single isolates, whereas heterogeneous populations were measured by well growth. It should also be noted as a general point that our experiments were not designed to identify the exact association between new variants and resistance phenotypes at all intermediate passage steps. Given the large number of total mutations, we focused only on those variants that were retained and fixed as resistance to CZA developed. It is possible that transient variants that did not fix contributed incrementally to resistance at intermediate stages, and our analysis would not have captured these variants. Similarly, we cannot rule out the possibility of epigenetic or other nongenetic mechanisms of resistance playing a role in the observed increased in MIC in our selection experiment [63,64].

We believe the findings in this work have important potential clinical implications. Firstly, clinicians should be aware of the possibility that hypermutator *P. aeruginosa* strains will be encountered in the hospital, and that these isolates may rapidly and predictably develop resistance under CZA selection. Secondly, in the course of such antibiotic selection, hypermutators may dynamically modify the set of available evolutionary pathways through mutational inactivation of classical resistance genes, leading to the evolution of novel resistance mechanisms for which antibiotics have not been developed. Alternatively, mutations leading to hypermutation may occur within the background of preexisting MexAB-OprM deficiency, which we found was relatively common among *P. aeruginosa* genomes in the NCBI Pathogen Database. These mechanisms may also confer cross resistance to other related antibiotics, including cefepime and C/T, as seen in this study. At present the exact clinical significance of these findings is unclear, but merits further investigation. Similar analyses performed with other agents that are used to treat *P. aeruginosa* in MMR-deficient hypermutator strains may reveal additional important insights. Given these findings, identification of hypermutator phenotypes in the clinical microbiology lab, either through direct phenotypic testing or through targeted gene amplification and sequencing, may in the future inform optimized treatment decisions based on these differences in behavior, but additional clarifying work on the role of hypermutation in the evolution of clinical resistance is needed.

There are a few important caveats that flow from the design of this study. The first is that the experiments were performed under in vitro conditions, where the predominant selection force was CZA concentration. In the natural context of in vivo infection, a number of other selection forces, including host defense mechanisms factor into the evolutionary equation, and it is possible that bacterial strains may face more stringent global purifying selection under these conditions that might limit the kind of mutational fixation we observed here. Secondly, the hypermutator experiments were performed in a single lab-strain genetic background, and it is possible that the resistance mechanisms uncovered will turn out to be dependent upon unique features of the lab strain genetic background and will not translate into general mechanisms conferring resistance in genetically dissimilar *P. aeruginosa* strains. Thirdly, although both elevated mutation rate and the ease with which non-beneficial mutations may fix (such as MexB W753R) may contribute to the accelerated evolution of resistance through alternative mechanisms in hypermutators, we are unable to distinguish the relative contributions of these 2 factors. As noted in the foregoing discussion, we believe that MutS-deficient hypermutation likely contributed the initial inactivating *mexB* mutation and then accelerated the search for alternative resistance mechanisms that was required by this mutation. These 2 events are thus inextricably linked in the actual process that occurred in our adaptive evolution experiments,

and we anticipate that similar behavior may be observed more generally in hypermutators. Separately, we observed a relatively high frequency of mutations that would be expected to inactivate MMR, BER ($\geq$2.3%), and MexAB-OprM ($\geq$3.9%) in publicly available *P. aeruginosa* genome data. These findings suggest that hypermutation may co-occur with a preexisting MexAB-OprM mutation leading to similar activation of alternative AMR resistance pathways as we observed in our work. In this context, we think it is important to note that in many studies describing MexAB-OprM mutations associated with antibiotic resistance, there is no functional corroboration of the mutant allele phenotypes through the construction of isogenic mutants or other methods. It is conceivable that some previously described MexAB-OprM variants in antibiotic-adapted strains may actually be inactivating, and the isolates may have developed resistance through less appreciated pathways due to the unavailability of MexAB-OprM.

While the MexVW mutations characterized above account for only a small proportion of the high level of resistance that evolved in the MMR-deficient isolates, they demonstrate conclusively that hypermutators can acquire antibiotic resistance through mechanisms not commonly used by WT isolates. Furthermore, the rarity of the observed mutations in public databases implies that the universe of mechanisms that can confer beta lactam resistance in *P. aeruginosa* is likely much larger than previously appreciated. The apparent ease with which hypermutators may switch to alternative resistance mechanisms for which antibiotics have not been developed may carry important therapeutic implications. Further study of the gene targets altered in these isolates may reveal additional unappreciated mechanisms, including potentially novel antibiotic targets.

## Materials and methods

### Bacterial strains

*P. aeruginosa* strain MPAO1 (MPAO1-WT) and its *mutS* transposon insertion mutant (MPAO1-*mutS*$^{Tn}$, isolate PW7149) were obtained from the U. WA Colin Manoil Lab Two Allele Library [65]. An increased spontaneous mutation rate in the MPAO1-*mutS*$^{Tn}$ isolate was confirmed experimentally by means of a rifampin reversion assay, as previously published [19]. *P. aeruginosa* isolate "PT" was a multidrug resistant, CAZ-susceptible, non-hypermutator clinical isolate from our institution, as previously reported [19]. All strains were susceptible to CZA at baseline (Fig 2).

### Experimental evolution procedure

In a previously published study [19], lineages of MPAO1-WT, MPAO1-*mutS*$^{Tn}$, and PT were serially passaged as follows. For each strain, the frozen stocks were inoculated on sheep blood agar (SBA, TSA with 5% sheep blood) (Remel, Lenexa, Kansas, United States of America) and incubated at 35°C for 18 to 24 h. A single colony was inoculated in 20 mL Luria–Bertani (LB) (BD Difco, Franklin Lakes, New Jersey, USA) broth and incubated at 35°C for 12 h in a shaking incubator at 250 RPM, and then diluted to an initial $OD_{600}$ of 0.05, corresponding to a CFU count of approximately $1.5 \times 10^8$ CFU/mL. This CFU count results in a relatively large initial critical population size (approximately $1.5 \times 10^7$ CFU) but was chosen to ensure robust survival of the population during passaging under antibiotic selection. Quadruplicate lineages from this stock were serially passaged on Luria–Bertani (LB) broth with a gradient of ceftazidime (Sigma-Aldrich, St. Louis, Missouri, USA) concentration in doubling dilutions ranging from 0.5 to 512 μg/mL with a fixed avibactam (Sigma-Aldrich, St. Louis, Missouri, USA) concentration of 4μg/mL. The passage interval was approximately 24 h, for a total of 20 passages. At each interval, cultures were aliquoted from the highest antibiotic concentration well with

an $OD_{600}$ >0.2 after background subtraction, diluted by 1:100, and re-inoculated in fresh media with antibiotic gradient. To confirm mutant stability and to obtain a measure of MIC in pure culture for a subset of representative isolates, susceptibility testing was performed by E-test (BioMerieux, Durham, North Carolina, USA) from 1 colony per lineage per passage after growth on SBA for 24 h following the E-test manufacturer's instructions. As a no-selection control, an identical scheme with quadruplicate lineages per strain was carried out on LB broth with no antibiotic and passaging after 1:100 dilution.

## Growth rate comparisons between MPAO1-WT, MPAO1-*mutS*$^{Tn}$

Three isolates from each lineage/passage shown in S4 Fig were streaked for isolation from glycerol stocks frozen at −80°C onto Remel TSA with 5% sheep blood agar and incubated at 35°C for 20 h. A quarter of a 10 mL inoculation loopful of colonies were collected and suspended in fresh LB broth. These cell mixtures were used to inoculate 200 mL LB to a target $OD_{600}$ of 0.01 as measured on a Biotek Cytation 5 instrument using the Gen5 Microplate Reader and Imager Software (Agilent Technologies, Santa Clara, California, USA). Hourly $OD_{600}$ measurements for each strain were collected at 37°C for 21 h and plotted using GraphPad Prism v.8 software (San Diego, California, USA).

## Susceptibility testing

All engineered mutants described as well as the MPAO1-WT underwent testing with an extended panel of antibacterials. Ampicillin, aztreonam, cephalothin, cefoxitin, ceftolozane/tazobactam (C/T), cefepime, CZA, ceftazidime, piperacillin, piperacillin/tazobactam, and meropenem MICs were obtained for *n* = 10 biological replicates with E-test for the MPAO1-WT, *mexV* -82:T>C, MexW E36K, and *mexV* -82:T>C + MexW E36K mutants. Testing with levofloxacin, ciprofloxacin, amikacin, tobramycin, and gentamicin was performed in *n* = 5 biological replicates in these mutants by the Kirby–Bauer method (Hardy Diagnostics Santa Maria, California, USA and BD Diagnostics, Franklin Lakes, New Jersey, USA). Additionally, the MPAO1-WT, MPAO1-WT + MexB W753R, MPAO1 1C12-1, MPAO1 1C12-1 + MexB W753R, *mexV* -82:T>C + MexW E36K, and *mexV* -82:T>C + MexW E36K + MexB W753R were tested with CZA, ceftazidime, and aztreonam by E-test in *n* = 6 biological replicates. All susceptibility testing was performed following CLSI guidelines [66]. Briefly, isolates were plated from frozen into LB agar plates and incubated for 24 h at 35°C, after which ≥5 colonies per isolate were resuspended in 0.85% NaCl to a turbidity of 0.5 McFarland and inoculated as a lawn using a sterile cotton-tip swab onto Mueller–Hinton agar plates (Hardy Diagnostics Santa Maria, California, USA and BD Diagnostics, Franklin Lakes, New Jersey, USA). After drying for 10 to 15 min, antibiotic test strips and discs were applied to the agar surface with tweezers. Plates were read after incubation for 18 to 20 h at 35°C.

A table displaying the discrepancies between measured CZA E-test MICs in a representative isolate per passage and the antibiotic concentration inhibiting growth in that passage during the antibiotic adaptation scheme is shown in S4 Data.

## Engineered mutant growth curves

Mutants and WT strains were streaked on LB plates from frozen stock and incubated overnight at 35°C. A single colony from each was resuspended in 5 mL of LB broth, incubated at 37°C for 18 h, and subsequently diluted to a final $OD_{600}$ = 0.001 (approximately $10^6$ CFU/mL) into fresh LB and LB broth supplemented with appropriate concentration of CZA. The 96-well plates containing 200 μL of the appropriate suspension per well were incubated at 37°C with continuous orbital shaking in a Cytation 5 plate reader for 36 h, and growth was measured as

OD$_{600}$ every 15 min. Growth curves were generated at 4 different concentrations: 0.5,1, 2, and 4 μg/mL of ceftazidime hydrate (Sigma Aldrich, St. Louis, Missouri, USA) with a fixed concentration of avibactam sodium at 4 μg/mL and in LB without CZA. Seven biological replicates per mutant and WT were used. Growth curves were plotted in R using the ggplot package.

## Whole-genome sequencing (WGS)

For each lineage, WGS was performed on 3 individual colony subcultures from passages straddling at least the clinically relevant range of CZA concentrations. Single colonies from the starting MPAO1-mutS$^{Tn}$ and MPAO1-WT stocks were subcultured for WGS as well. DNA extraction was performed with easyMAG (Biomerieux, Durham, North Carolina, USA) or DNeasy Blood and Tissue kit (QIAGEN, Hilden, Germany). DNA libraries were prepared with either Nextera XT, Nextera Flex (Illumina, San Diego, California, USA), New England Biolabs NEBNext or iGenomX Riptide (iGenomX, Carlsbad, California, USA) [67] methods. Sequencing was done using 3 Illumina platforms: MiSeq with 300-bp paired-end reads, iSeq 100, or HiSeq 2500 or NextSeq 550 with 150-bp paired-end reads. Reads were aligned to PAO1 GenBank reference (RefSeq accession number NC_002516.2) for the MPAO1-WT and MPAO1-*mutS$^{Tn}$*; or our de novo draft assembly for the PT strain [19], using BWA v0.7.17 (https://github.com/lh3/bwa) with default parameters, and filtered for reads with "proper pairs" using Samtools v. 1.6 (https://github.com/samtools/samtools). In order to have independent confirmation of SNVs found in the terminal isolates, they were additionally sequenced using Nextera XT or NEBNext Ultra II library preparation. For this purpose, terminal isolates were defined as isolates from the latest passage of CZA adaptation that were recoverable for sequencing (S11 Table).

One representative isolate from each of 4 lineages per strain (MPAO1-WT and MPAO1-*mutS$^{Tn}$*), passaged without antibiotic selection was sequenced after 11 passages. DNA extraction was performed with DNeasy Blood and Tissue kit (QIAGEN, Hilden, Germany) and DNA libraries were prepared with Nextera Flex. WGS was performed with an Illumina iSeq instrument (150-bp PE reads). Downstream variant analysis was carried out in an identical way as with the CZA-adapted isolates. A list of variants called in these isolates can be found in the S5 Data.

## Quality control and variant calling

A total of 376 genomic DNA libraries from lineages passaged under CAZ selection were initially sequenced. Libraries with fewer than 1,000,000 reads (iGenomX) or fewer than 200,000 reads (Nextera) were excluded from analysis. These different cutoffs were used to account for the inherent difference in homogeneity of coverage between the library preparation methods, as well as the fact that Nextera libraries were sequenced with 300-bp paired-end reads, as opposed to 150-bp paired-end reads for the iGenomX approach. Quality control was performed with Picard Tools v2.25.0 (https://broadinstitute.github.io/picard/) using the CollectWgsMetrics and CollectAlignmentSummaryMetrics tools, and samples with fewer than 90% of bases achieving at least 5X coverage or with <98% reads aligned to reference were excluded from further analysis. A total of 303 genomes that met the above criteria were used for all further analysis.

Variant calling was done with Freebayes v. 1.1.0 (https://github.com/ekg/freebayes) against the corresponding reference using haploid mode, a minimum allele fraction >0.8 for the iGenomiX libraries (>0.5 for terminal isolates independently sequenced with Nextera XT or NEBNext Ultra II libraries, as described below), minimum read count ≥5, mapping quality >100, and standard quality filters (—ploidy 1—min-alternate-fraction 0.8—min-alternate-count 5–0. The -0 option indicates "standard" freebayes filters: -m 30 (minimum mapping quality 30), -q

30 (minimum base quality 30)—R 0 (min supporting sum of qualities 0), -S 0 (genotype variant threshold 0)). A known highly variable Pf4 phage island was observed in our sequencing data (GenBank PAO1 reference positions approximately 721,400 to 740,400) that resulted in many called variants. Variants called within this region were not included in the analysis.

Variants were analyzed using the vcftools v.0.1.16 (https://github.com/vcftools/vcftools) and R v. 4.0 (https://www.r-project.org/) softwares. The starting MPAO1-WT and MPAO1-*mutS*[Tn] strains share 55 common variants relative to the PAO1 reference, which were subtracted from analysis. For the MPAO1-*mutS*[Tn] lineages, an additional set of 54 variants were present by the end of the first day of passage, denoted as "early variants" throughout the text. These included 33 variants present in the MPAO1-*mutS*[Tn] isolate starting stock culture plus an additional 21 variants that were present by the end of the first passage. The latter were based on sequencing 3 isolates each from passage 1 of lineages 2A and 2D (S1 Data). We assessed for large deletions by analyzing read counts aligning to each annotated gene in the PAO1 reference genome, normalized by gene length, sample, and run coverage. Regions with normalized read count values below 0.1 supported in terminal isolates were considered as signatures of true large deletions and were included in the analysis.

Some genome regions had consistently low coverage in the iGenomX libraries, resulting in difficulty differentiating some true variants from technical artifact. To confirm fixed variants identified in the iGenomX libraries, genomic DNA libraries from the terminal isolates of each lineage were additionally sequenced with a second library preparation method (Nextera XT or NEBNext Ultra II). We focused our analysis on fixed variants, which were defined as variants present in (1) at least 2 different passages or the final passage within a single lineage; and (2) present in the terminal isolates independently sequenced with Nextera XT or NEBNext Ultra II for the lineage. The characteristics of each fixed variant were analyzed with respect to passage of appearance, strain, and CZA E-test MIC using the tidyverse (https://www.tidyverse.org/) R package. Genomic analysis detected possible cross-contamination of MPAO1-*mutS*[Tn] lineage 2B into 2C at passage number 4, affecting 12 isolates sequenced in this lineage. Though the resulting lineage 2C then was passaged independently and evolved a 6 doubling dilution increase in CZA MIC over these passages, we chose to remove it from aggregate genomic analysis at this stage as a conservative approach.

Variant annotation was done with snpEff v. 4.3 (https://pcingola.github.io/SnpEff/) with default parameters, using local PAO1 and PT reference databases generated from Genbank (.gbk) files, using the "java -jar snpEff.jar build" command. The NCBI PAO1 reference (RefSeq accession number NC_002516.2) was used for MPAO1-WT and MPAO1-*mutS*[Tn], while the.gbk file generated by the Prokka software (https://github.com/tseemann/prokka) was used for the PT lineages. Variants were then extracted and aggregated for analysis with snpSift after transformation to 1 entry per line (command: $SNPEFF_HOME/scripts/vcfEffOnePerLine.pl | java -Xmx10g -jar $SNPSIFT_JAR extractFields—CHROM POS REF ALT "ANN[*].ALLELE" "ANN[*].EFFECT" "ANN[*].IMPACT:" "ANN[*].FEATURE" "ANN[*].FEATUREID" "ANN[*].CDS_POS" "ANN[*].CDS_LEN" "ANN[*].AA_POS" "ANN[*].AA_LEN" "ANN[*].DISTANCE" "ANN[*].ERRORS").

## Allele frequency, variant impact, and conservation analysis

The allele frequency for a fixed variant in a given passage was calculated as the percentage of isolates possessing the variant over the number of isolates sequenced in that passage. Calculated conservation, Jensen–Shannon divergence score, breadth, and variant impact scores were calculated as follows. All protein sequences were downloaded from the 100-genome *Pseudomonas* genome ortholog database (https://pseudomonas.com/rbbh/pairs) (S6 Data).

MAFFT v7.475 was used to generate multiple sequence alignments for each of the PAO1 reference genome proteins with their respective orthologs. A previously described algorithm [48] was then used to assign each amino acid residue a score based on the Jensen–Shannon divergence and to calculate a percentile across all other residues in the same protein. The "breadth" for each protein was defined as the percentage of *Pseudomonas* isolates in this set of 100 genomes containing an ortholog of the given protein. The "Variant impact" parameter was collected from the SnpEff output, defined roughly as: High impact = frameshift or nonsense variants; Moderate impact = missense variants or in-frame deletions; Low impact = synonymous variants; Modifier = noncoding variants.

## Gene ontology enrichment analysis

Among 139 fixed variants in MPAO1-MutSTn, we identified 78 distinct genes with at least 1 non-synonymous variant. To perform functional annotation, the GO terms for the complete set of PAO1 loci were obtained from the *Pseudomonas* genome database [40]. A total of 4,042 PAO1 loci were annotated with GO terms and were suitable for analysis. We then utilized the topGO v2.46.0 R package [68] to perform an enrichment analysis using the weight01 algorithm with the set of 78 mutated genes. Enriched GO terms with a 1-tailed Fisher's exact test $p < 0.05$ are shown on S2 Fig.

## Growth rate comparisons of strains in the adaptive evolution experiment

Each isolate was streaked for isolation on TSA with 5% sheep blood agar plates (Thermo Fisher Scientific, Waltham, Massachusetts, USA), incubated at 35°C for 20 h, and then collected and suspended in fresh Luria–Bertani (LB) broth. This suspension was used to inoculate 200 μL LB to a target $OD_{600}$ of 0.01 as measured on a Biotek Cytation instrument using the Gen5 Microplate Reader and Imager Software. Growth curves for each strain were measured in replicate for 23 h with shaking at 180 RPM.

## RNA sequencing

Twenty-two isolates from the in-vitro evolution experiment were chosen for RNA-seq analysis. The set included $2 \times 0$, ancestor strain of the hypermutator lineages from the adaptive evolution experiment, and all terminal and intermediate isolates listed in S3 Data for each lineage (2A, 2B, and 2D). The ancestor of the WT lineages in the adaptive evolution experiment, $1 \times 0$, was also chosen, along with all terminal isolates in 3 WT lineages (1A, 1C, 1D).

To determine the mid-log point for cell harvesting, each isolate was streaked for isolation onto TSA with 5% sheep blood agar plates (Remel, Thermo Fisher Scientific, Waltham, Massachusetts, USA) and incubated at 35°C for 20 h, or 24 h for MPAO1-$mutS^{Tn}$ 2B7-1. Cells were resuspended in 10 mL of LB broth to an initial $OD_{600}$ of 0.01 and incubated in a vented 50 mL tube at 37°C with shaking at 220 RPM for 12 h, with $OD_{600}$ readings obtained every hour. Mid-log growth was calculated from half the stationary phase $OD_{600}$. To facilitate the collection of cells from each lineage for a high-throughput downstream RNA-seq experiment, a universal time point for mid-log was approximated for the MPAO1-$mutS^{Tn}$ group (5 h) and the MAPO1-WT group (5.5 h).

For cell harvest for RNA extraction, 10 mL liquid subcultures were started from plates as above with a starting $OD_{600}$ of 0.01 and grown in a vented 50 mL tube at 37°C with shaking at 220 RPM. At the mid-log time points, a final OD measurement was taken and 1.5 mL of culture was mixed with 3 mL of RNA-Protect Reagent (Qiagen, Germantown, Maryland, USA). Cell harvest was conducted as per the manufacturer's instructions. Briefly, the culture/RNA-protect mixture was vortexed for 5 s, incubated at room temperature for 5 min, then

centrifuged at 4,000 rcf for 10 min. The supernatant was decanted, and the cell pellet was allowed to dry. Each pellet was stored at −80°C until RNA isolation.

For RNA extraction, thawed cell pellets were resuspended in 400 μL of RNase-free molecular grade water and immediately mixed with 550 μL of Binding Buffer and bead mix from the MagMAX Viral/Pathogen Nucleic Acid Isolation Kit (Applied Biosystems, Foster City, California, USA). All required plates were prepared according to manufacturer's instructions and loaded into a KingFisher Flex Purification System (Thermo Fisher Scientific, Waltham, Massachusetts, USA). Total RNA was isolated from resuspended pellets using the standard MagMax Viral/Pathogen Nucleic Acid Isolation Kit protocol but modified to include a DNase treatment. For this, 3 extra plates were prepared and loaded into the KingFisher. One was a DNase plate that contained 98 μL of 1X Turbo DNase Buffer (Invitrogen, Carlsbad, California, USA) mixed with 2 μL of DNase (Invitrogen, Carlsbad, California, USA) per well, to be used per sample. Another plate was prepared with 1 mL of Wash Buffer (Applied Biosystems, Foster City, California, USA) per well, per sample, and the other contained 1 mL of fresh 80% ethanol per well, per sample. The 2 latter plates were used for pre-DNase washing of each sample. RNA was eluted in 100 μL volumes of Elution Buffer (Applied Biosystems, Foster City, California, USA). RNA concentration, quality, and purity were checked with a NanoDrop One Microvolume UV-Vis Spectrophotometer (Thermo Scientific, Waltham, Massachusetts, USA) and a 5200 Tapestation (Agilent, Santa Clara, California, USA).

All resulting isolated RNA selected as input for ribosomal RNA depletion and library preparation had A260/280 ratios ranging from 2.05 to 2.41 and RNA integrity numbers equal to or above 6. Depletion was performed with the NEBNext rRNA Depletion Kit for bacteria (New England Biolabs, Ipswich, Massachusetts, USA) following the manufacturer's protocol without modification. RNA libraries were then prepared using the NEBNext Ultra II Directional RNA Library Prep Kit for Illumina and NebNext Multiplex Oligos for Illumina (New England Biolabs, Ipswich, Massachusetts, USA). Both protocols were performed on an epMotion 5075 (Eppendorf, Hamburg, Germany).

Final cDNA libraries were quantified using a Qubit 4.0 Fluorometer (Invitrogen, Waltham, Massachusetts, USA), and average fragment lengths were determined using the Agilent Fragment Analyzer (Agilent Technologies, Santa Clara, California, USA). Samples for which libraries had a measured dsDNA concentration <5 ng/dL or average peak size <300 bp were deemed inadequate for analysis and re-harvested as described above. Final libraries were diluted to 4 nM for single-end sequencing on an Illumina NextSeq using the NextSeq 500/550 High Output cartridge v2.5 75-cycle kit (Illumina, San Diego, California, USA), obtaining a total of 2 to 13 million reads per sample (S3 Data).

Following sequencing, reads were preprocessed with Trimmomatic Version 0.39 using the following parameters: universal Illumina adaptors were trimmed using the TruSeq3 single end mode, simple clip threshold of 10 and adapter seed mismatches of 2. Surviving reads were aligned to the *P. aeruginosa* PAO1 reference genome (available from https://www.pseudomonas.com/) using Burrows–Wheeler Aligner version 0.7.17. Bam alignment files and the PAO1 reference GFF file (https://www.pseudomonas.com/) were used to generate read counts for each sample with HTseq v0.11.4, using the htseq-count -s reverse -m union flags. In parallel, Samtools v1.11 and Picard CollectRnaSeqMetrics (Picard tools v2.25.0) were used to obtain QC metrics. One sample (MPAO1-*mutS*^Tn 2D3 replicate B) was deemed to be an outlier based on a cutoff of <70% bases aligned to the correct strand and a 5′ to 3′ bias >0.8 and was removed from downstream analysis.

To quantify gene expression, the DESeq2 v1.30.1 package (https://bioconductor.org/packages/release/bioc/html/DESeq2.html) was used to obtain rlog-normalized read counts for all samples (DESeq2 rlog function) that were then used for PCA plot (DESeq2 plotPCA function). To analyze differential gene expression, the DEseq2 (DESeq function with formula ~

lineage) was used to obtain log2-fold estimates of expression and adjusted false discovery rates for each evolved isolate compared to its respective ancestor. These values were shrunk using the lfcShrink function with type = apeglm. Each differential gene expression table was cross-referenced with the per-gene metadata available from the *Pseudomonas* genome database (https://www.pseudomonas.com/) as well as the genomic variant, passage, and CZA MIC data from the in vitro evolution experiment. Volcano plots and linear plots were constructed using the ggplot2 package (https://ggplot2.tidyverse.org/). Venn diagrams showing the common up/down-regulated genes per evolved terminal isolate per lineage were generated with the ggVenn R package (https://github.com/yanlinlin82/ggvenn). For heatmap construction, the subset of the top 750 genes with greatest variance was used as per defaults of the genefilter package (https://bioconductor.org/packages/release/bioc/html/genefilter.html). For display, a Euclidean distance matrix of the rlog-normalized expression for each gene and isolate was clustered row-wise using hierarchical clustering and plotted/annotated using the ComplexHeatmap package (https://github.com/jokergoo/ComplexHeatmap).

## Construction of the mutations in wild-type *P. aeruginosa* MPAO1

In order to engineer mutations related to the MexVW efflux pump into the WT MPAO1, we used a previously published 2-step allelic exchange method [69]. First, we constructed 2 single mutants. The first contained the single point mutation in *mexW* (4904752 G>A) to generate the E36K substitution in the translated MexW protein, hereafter called "*mexW* E36K" for brevity. The second mutation (T>C) was introduced at position 4,903,384 upstream of *mexV* into the WT MPAO1 background. These 2 mutations were then combined to obtain a double mutant. To construct the allelic exchange vector pEX18Gm (hereafter: vector), we first synthesized the mutant allele gene fragment (hereafter: fragment) (gBlocks HiFi Gene Fragments, Integrated DNA Technologies) containing the desired mutation with 500-bp upstream and 500-bp downstream regions. In order to clone the fragment into the exchange vector, we added XbaI and HindIII restriction sites at the 5′ and 3′ ends of the fragment, respectively. The exchange vector and the synthesized gene fragments were digested with XbaI and HindIII (New England Biolabs) and subsequently ligated and transformed into the NEB 5-alpha Competent *E. coli* (New England Biolabs). The insertion of the mutant allele into pEX18Gm and sequence were verified by Sanger sequencing using M13F and M13R universal primers. We next transformed the engineered exchange vector into *E. coli* S17.1 (λ pir+) and performed biparental mating with WT *P. aeruginosa* MPAO1. Single recombinant mutants were selected on Vogel-Bonner minimal medium containing 60 μg/mL gentamicin. Double recombinant mutants were selected on no salt LB agar plates containing 15% (w/v) sucrose. The colonies were screened for the point mutation by PCR amplification (Phusion High-Fidelity DNA polymerase, Thermo Fisher Scientific) and Sanger sequenced using appropriate primers (S12 Table). For the construction of the double mutant (position 4903384 T>C and *mexW* E36K), the pEX18Gm:: *mexW* E36K construct was conjugationally transferred to the *P. aeruginosa* MPAO1 position 4903384 T>C single point mutant and selected single and double crossover mutants colonies as described above.

The 2-step allelic exchange method using the pEX18Gm vector with few modifications was used to construct the *mexB* (475447 T>C) mutation yielding the W753R substitution in the translated protein, in the WT MPAO1, MPAO1 *mexV* -82:T>C + *mexW* E36K, and MPAO1-1C12-1 strain backgrounds. First, we synthesized the mutant allele gene fragment (gBlocks HiFi Gene Fragments) containing the desired mutation with 400-bp upstream and 400-bp downstream regions and cloned into allelic exchange vector pEX18Gm as described above. The allelic exchange vector pEX18Gm:: *mexB* W753R was transformed into *E. coli* S17.1 (λ pir+) and further used to transfer the vector pEX18Gm:: *mexB* W753R into different

*P. aeruginosa* strains through biparental matting and selected single and double recombinants as described above. Single recombinant merodiploid for MPAO1 1C12-1 were selected on *Pseudomonas* isolation agar (BD Biosciences) containing 60 μg/mL gentamicin. Double recombinant mutants were screened for the desired point mutation by PCR amplification (Phusion High-Fidelity DNA polymerase, Thermo Fisher Scientific) and Sanger sequencing using appropriate primers (S12 Table).

All of the constructed mutants above underwent whole-genome sequencing to verify the desired mutations and absence of any additional mutations. Briefly, individual colonies from each mutant were selected after overnight growth on LB agar, DNA extracted and libraries prepared using the Nextera Flex/Illumina DNA (Illumina, San Diego, California, USA) library preparation kit, with sequencing performed on an iSeq 100 platform with 150-bp PE reads. Reads were aligned to the PAO1 GenBank reference (RefSeq accession number NC_002516.2) with an additional contig for the pEX18Gm cloning vector, using BWA software v0.7.17 (https://github.com/lh3/bwa) with default parameters. The alignment was filtered for reads with "proper pairs" using Samtools v. 1.6 (https://github.com/samtools/samtools). Variant calling was done with Freebayes v. 1.1.0 (https://github.com/ekg/freebayes) against the corresponding reference, using haploid mode, a minimum allele fraction >0.8, minimum alternate count >5, mapping quality >100, and standard quality filters. Called variants were inspected manually in the Broad Institute Integrative Genomics Viewer (IGV) [70,71].

## BLAST search for MexVW mutations fixed in MPAO1-*mutS*$^{Tn}$ lineages in NCBI Pathogens Database

FASTA nucleotide sequences for 7,493 *P. aeruginosa* genomes were downloaded from the NCBI Pathogens Database (accessed on May 14, 2021). All FASTA files were annotated with Prokka v1.14.6 (https://github.com/tseemann/prokka) and local BLASTn and BLASTp databases were created with the NCBI BLAST [72] makeblastdb command. A BLASTn search was conducted using as queries the intergenic region between PA4373-*mexV* as well as the complete nucleotide sequence of *mexW* (PA4375). Similarly, the protein sequence of MexW was used as a query for a BLASTp search in the local protein database. A concatenated FASTA for all hits with percent identity > 90% and E value < $10^{-6}$ was used to generate MAFFT multiple sequence alignments. To find and quantify the variants of interest, alphabet frequencies were calculated around the residues of interest, using the Biostrings v0.32.0 (https://bioconductor.org/packages/release/bioc/html/Biostrings.html) R package.

## Identification of putative hypermutators in NCBI Pathogen Detection Database

To identify putative hypermutators, we searched for genomes containing highly disruptive mutations in MMR and BER proteins. We used BLASTp v2.13.0 to identify sequences homologous to *P. aeruginosa* PAO1 MutS (NP_252310.1), MutL (NP_253633.1), UvrD (NP_254130.1), MutT (NP_253090.1), MutY (NP_253834.1), and MutM (NP_249048.1) proteins in *P. aeruginosa* genome assemblies in the NCBI Pathogen database (accessed 5/14/2022, *n* = 6,877 genomes with available assemblies and metadata). We retrieved all matches with e-value <$10^{-4}$ and percentage identity >94% and then excluded assemblies with >1 BLASTp hit overlapping > 95% of the full-length PAO1 protein. Three assemblies were found to have potential duplications of: UvrD (GCA_011466835.1), MutY (GCA_013114955.1), and MutM (GCA_001180385.1). Assemblies with multiple partial matches per analyzed protein were likely due to frameshifts and nonsense mutations resulting in multiple ORFs annotated per homologous protein by prokka. For variant calling, we included assemblies with at least 1

BLASTp hit for all these 6 proteins ($n$ = 6,805). These were used as input for snippy v4.4.1. with—contigs option to generate synthetic reads that were aligned back to the *P. aeruginosa* PAO1 genome (NC_002516.2) to generate variant calls. Variants were annotated with SnpEff v5.1d. The variants within the coding sequences of *mutS*, *mutL*, *uvrD*, *mutT*, *mutY*, or *mutM* were subsequently aggregated and analyzed using the tidyverse Bioconductor package in R 4.13. We calculated the percentage of assemblies in the database with at least 1 highly disruptive (stop gained, start lost, or frameshifting) variant out of 6,805 assemblies. For a highly conservative estimate to account for potential multiple sampling of clonal lineages in the dataset, we assumed each unique variant to represent a single clonal lineage and divided this number by the same denominator of 6,805 assemblies as used above.

## Assessment of putative MexAB-OprM inactivation in NCBI Pathogen Detection Database

Similar to the approach described above, we searched for MexAB-OprM inactivation. As above, in order to identify assemblies with duplications or no identifiable homologues, we used BLASTp to align amino acid sequences of PAO1 MexA (NP_249116.1), MexB (NP_249117.1), and OprM (NP_249118.1) to prokka annotated proteins from 6,877 assemblies with <500 contigs and associated metadata in the NCBI Pathogen Detection database, using the same search cutoffs (e-value $<10^{-4}$ and % identity >94%). We identified 1 assembly (GCA_000520315.1) containing a putative MexA duplication, which was excluded from analysis. Assemblies with at least 1 BLASTp hit for MexA, MexB, and OprM ($n$ = 6,820) were used as input for snippy and SnpEff to perform variant calling and annotation as described above. Variants were analyzed with the tidyverse Bioconductor package in R 4.13. We calculated the percentage of assemblies in the database with at least 1 highly disruptive (stop gained, start lost, or frameshifting) variant out of the 6,820 assemblies. As above, for a conservative estimate to account for potential multiple sampling of clonal lineages in the dataset, we assumed each unique variant to represent a single clonal lineage and divided this number by the same denominator of 6,820 assemblies as above.

## Statistical analysis

The proportion of transitions / transversions in MPAO1-WT versus MPAO1-*mutS*$^{Tn}$ was analyzed using a 2-tailed Fisher's exact test using the GraphPad software (https://www.graphpad.com/quickcalcs/contingency1.cfm). Other statistical analysis of differential gene expression was performed with DESeq2 v1.26.0. The Wilcoxon/Mann–Whitney test was performed with the R built-in function Wilcox-test.

## Figures and table construction

Variant metadata and variant features were analyzed in R software using the tidyverse package (https://www.tidyverse.org/). Figures were generated using the R ggplot2 v3.3.3 (https://ggplot2.tidyverse.org/), ComplexHeatmap v2.2.0 (https://github.com/jokergoo/ComplexHeatmap), EulerR v6.1.0 (https://github.com/jolars/eulerr), and patchwork v1.1.1 (https://github.com/thomasp85/patchwork) packages.

## Ethics statement

The work presented in this manuscript involved only laboratory *P. aeruginosa* strains PAO1 and MPAO1 and a single de-identified clinical *P. aeruginosa* strain (PT). As such, this work was exempt from NIH IRB review.

## Supporting information

**S1 Fig. Proportions and types of fixed variants acquired during CZA adaptation for each genotype and lineage.** The total number of fixed variants acquired is indicated in parentheses. The underlying data to generate this figure can be found in S2 Data.
(PDF)

**S2 Fig. Gene Ontology enrichment analysis of targets of fixed mutations in *P. aeruginosa* hypermutator lineages.** Significantly enriched GO terms (1-tailed Fisher's exact test $p < 0.05$) for each GO domain are displayed in order of decreasing $p$-value. The x-axis represents the Log2 (observed/expected) ratio. The underlying data to generate this figure can be found in S2 Data. CC, cellular component.
(EPS)

**S3 Fig. Characteristics of fixed variants associated with genes mutated more than once across MPAO1-*mutS*$^{Tn}$ lineages.** Subset from Fig 2 showing variants that became fixed in at least 2 of the 3 MPAO1-*mutS*$^{Tn}$ lineages (2A, 2B, and 2D), plotted with CZA MIC (E-test) and passage number (tick marks). A filled tile indicates the presence of the variant at the corresponding passage and the shade of blue represents the proportion of isolates in a given passage carrying the corresponding variant, as defined in the allele frequency legend bar at the right. Sidebars are defined identically as in the Fig 2 legend. The underlying data to generate this figure can be found in S2 Data. FS, frameshift variant.
(EPS)

**S4 Fig. MPAO1-WT and MPAO1-*mutS*$^{Tn}$ evolved lineages do not show gross growth deficits in LB broth.** Growth curves of isolates from MPAO1-WT lineage 1A and MPAO1-*mutS*$^{Tn}$ lineages 2A, 2B, and 2D (LB broth at 37˚C). The passage number for each curve is indicated next to the lineage name in each legend. Experiments were repeated in triplicate, and time points represent the average $OD_{600}$ with whiskers indicating the range. The underlying data to generate this figure can be found in S2 Data.
(TIF)

**S5 Fig. Number of differentially expressed genes in MPAO1-WT and MPAO1-*mutS*$^{Tn}$ isolates quantified by RNA-seq.** The number of up-regulated and down-regulated genes ($|LFC| > 1$, $p_{adj} < 0.01$) are depicted in blue and red, respectively, for the lineage and passage indicated. LFC was calculated relative to ancestral isolates for each of the lineages. The genotype/lineage and passage are shown in the x-axis: genotype (1 = WT, 2 = MPAO1-*mutS*$^{Tn}$); lineage (A-D); passage number. Using a cutoff FDR $< 0.01$ and $|LFC| \geq 1$, we observed a greater mean number of DEGs in the MPAO1-*mutS*$^{Tn}$ terminal isolates as compared to the terminal WT isolates, but this difference was not statistically significant (690 MPAO1-*mutS*$^{Tn}$ DEGs vs. 515 MPAO1-WT DEGs total, Wilcoxon rank sum test $p = 0.4$; 317 MPAO1-*mutS*$^{Tn}$ genes vs. 241 MPAO1-WT genes up-regulated, $p = 0.2$, and 373 MPAO1-*mutS*$^{Tn}$ genes vs. 274 MPAO1-WT genes down-regulated, $p = 0.4$). The underlying data to generate this figure can be found in S2 Data.
(EPS)

**S6 Fig. Principal component analysis of the MPAO1-WT and MPAO1-*mutS*$^{Tn}$ transcriptomes demonstrates global differences.** The plots are based on **(A)**, the rlog-normalized read counts or **(B)**, LFC compared to the corresponding lineage ancestor for each coding sequence. The numbers correspond to passages under CZA selection. The underlying data to generate this figure can be found in S2 Data.
(EPS)

**S7 Fig. No difference in expression levels of major resistance genes between MPAO1-WT and MPAO1-*mutS*$^{Tn}$ ancestors.** Inverted volcano plot displaying the LFC in MPAO1-WT ancestor vs. MPAO1-*mutS*$^{Tn}$ ancestor. Genes previously described in the literature to be associated with antibiotic resistance are labeled in blue, with the 6 major beta-lactam resistance determinants labeled in red. The horizontal line represents a cutoff for p-adjust value of 0.01. The underlying data to generate this figure can be found in S2 Data.
(EPS)

**S8 Fig. Shared DEGs in MPAO1-*mutS*$^{Tn}$ lineages following CZA selection.** Number of DEGs shared by the 3 MPAO1-*mutS*$^{Tn}$ lineages are shown, either up-regulated (left, LFC<-1, p-adj <0.01) or down-regulated (right LFC>1, p-adj <0.01) in the terminal CZA-adapted isolates. The underlying data to generate this figure can be found in S2 Data.
(EPS)

**S9 Fig. Engineered mutant (*mexV* -82:T>C, MexW E36K and double mutant) E-test MICs with β-lactam antibiotics in addition to those shown in Fig 4.** Each box represents $n$ = 10 biological replicates. The boxes display the median with the lower and upper hinges corresponding to the first and third quartiles. Brackets with asterisks above the plot indicate statistical significance (Wilcoxon 2-sided $p$-value < 0.05). The underlying data to generate this figure can be found in S2 Data.
(EPS)

**S10 Fig. Colonies observed inside the zones of inhibition in disk diffusion susceptibility tests in MPAO1-*mutS*$^{Tn}$ isolates.** Representative photos of Kirby–Bauer susceptibility testing with ATM, IPM, MEM, and piperacillin-tazobactam (TZP) in the MPAO1-WT ancestor (1X), MPAO1-*mutS*$^{Tn}$ ancestor (2X), and 2 CZA-resistant MPAO1-*mutS*$^{Tn}$ isolates (lineage 2A passage 12 and lineage 2B passage 7). Pictures display readings at both 16 h, as recommended per CLSI guidelines, as well as at 24 h of incubation at 37˚C. Scattered colonies can be observed inside some zones of inhibition at 16 h, particularly in the MPAO1-*mutS*$^{Tn}$ isolates and more markedly in the evolved CZA-resistant isolates. ATM, aztreonam; IPM, imipenem; MEM, meropenem; TZP, piperacillin-tazobactam.
(PNG)

**S11 Fig. Effect of MexB W753R substitution on aztreonam, ceftazidime, and ceftazidime/avibactam E-test MICs.** Each box represents $n$ = 6 biological replicates. The boxes display the median with the lower and upper hinges corresponding to the first and third quartiles. Brackets above the plot indicate statistical significance (Wilcoxon 2-sided $p$-value < 0.05) between MexB W753R mutants and their respective background. The underlying data to generate this figure can be found in S2 Data.
(EPS)

**S1 Table. General characteristics of WGS libraries and fixed variants.** Number of isolates and fixed variants analyzed per strain and general WGS sample statistics. The graphical representation of the fixed variants acquired by genotype described here is shown in the bar plots in Fig 1. The numbers in parentheses indicate the percentage out of all variants in the given genotype.
(DOCX)

**S2 Table. General characteristics of isolate PT reference assembly.**
(DOCX)

**S3 Table. Fixed variants acquired in the PT lineages.** CZA MICs are expressed in µg/mL. NA = passage was not sequenced.
(DOCX)

**S4 Table. Variants in genes mutated in MPAO1-*mutS$^{Tn}$* lineages that were passaged with and without CZA.** Three lineages (A, B, and C; left) passaged without CZA, compared to the 3 lineages passaged with CZA (2A, 2B, 2D; right), and also described in Fig 2. The numbering following the CZA lineage indicates the passage number in which the variant was identified, followed by the isolate number (e.g., 2A4-2 indicates lineage 2A, passage 4, isolate 2). A plus sign indicates presence of the variant. Coordinate positions are given with respect to the PAO1 reference chromosome. The MPAO1-WT lineages did not acquire any variants in shared targets. FS, frameshift mutation.
(DOCX)

**S5 Table. Extended susceptibility testing results for the engineered mutants.** Five biological replicates (A–E) from each of the 3 mutants that underwent extended antibiotic testing by the Kirby–Bauer method as described. Results are given as zone diameters in mm and corresponding CLSI breakpoint susceptibility interpretations are given in parentheses. AMK, Amikacin; CI, Ciprofloxacin; GEN, Gentamycin; TOB, Tobramycin.
(DOCX)

**S6 Table. Antimicrobial susceptibility testing of MPAO1-*mutS$^{Tn}$* ancestor and terminal isolates.** Susceptibility testing in the hypermutators was complicated by the appearance of satellite colonies within the inner zones. We made the assumption that the outer zone reflected resistance conferred by the genomic background of the hypermutator and that the inner zone satellite colonies represented mutants able to resist higher concentrations of antibiotic. In order to determine the outer zone diameter, we visually identified the zone of confluent bacterial growth and measured the corresponding diameter. We then measured the inner diameter by measuring the distance between the disk edge and the closest satellite colony and calculated the diameter using this distance. Diameters are shown in mm except for C/T, where testing was performed by E-test and results are show in µg/mL. Values within parentheses correspond to the CLSI interpretation of the zone diameter. NA = no satellite colonies seen within zone. Letter in parenthesis indicates biological replicate and R1 and R2 represent technical replicates. Pip-Tazo = Piperacillin-Tazobactam.
(DOCX)

**S7 Table. *P*-values of 2-sided Wilcoxon tests performed between the E-test MICs of engineered mutants (MPAO1 *mexV* -82:T>C, MPAO1 MexW E36K, and MPAO1 *mexV* -82: T>C + MexW E36K) and MPAO1-WT for the indicated agents.**
(DOCX)

**S8 Table. Consensus sequences and nucleotide and amino acid frequency tables in proximity to PA4373-mexV (4903384 T>C) and MexW E36K variants in 7,493 genomes from the NCBI pathogen database.** Orthologous regions neighboring 2 mutations in the MexVW operon were identified in a set of 7,493 *P. aeruginosa* genomes from the NCBI Pathogen Detection database. **(a)** Summary of the nucleotide sequence alignment in proximity to PA4373-*mexV* +/- 10 bases. (b) Amino acid sequence surrounding MexW E36K with the corresponding residue highlighted in orange background.
(DOCX)

**S9 Table. Highly disruptive variants in MMR and BER proteins in the NCBI Pathogen Detection Database.** Percentage calculations in "Number of assemblies" column represent the

number of assemblies containing variants in the given gene divided by the number of assemblies with at least 1 high-similarity BLASTp match for all proteins in the DNA repair system (*n* = 6,805). The percentage calculations in "Number of variants" column represent the number of unique variants divided by the same denominator above. This latter percentage represents a conservative estimate of the variant frequency adjusted for clonality with the assumption that all assemblies containing a given variant are clonal.
(DOCX)

**S10 Table. Highly disruptive variants in MexAB-OpM in the NCBI Pathogen Detection Database.** Percentage calculations in the "Number of assemblies" column represent the number of assemblies containing variants in the given gene divided by the number of assemblies with at least 1 high-similarity BLASTp match for all proteins in the system (*n* = 6,820). The percentage calculations in "Number of variants" column represent the number of unique variants divided by the same denominator above. This latter percentage represents a conservative estimate of the variant frequency adjusted for clonality with the assumption that all assemblies containing a given variant are clonal.
(DOCX)

**S11 Table. Passages from which terminal isolates were derived, for WGS and RNA-Seq analyses.** Passages indicated as "N/A" were not utilized for RNA-Seq analysis.
(DOCX)

**S12 Table. Bacterial strains and plasmids used in this work.**
(DOCX)

**S1 Data. Annotation of variants present in MPAO1-WT relative to PAO1 (sheet "Variants in MPAO1-WT vs. PAO1") and variants present in MPAO1-*mutS^{Tn}* relative to MPAO1-WT, separated into (1) those variants present in the starting stock (sheet "Muts-Tn starting stock variants"), and (2) those additional variants present at the end of the first passage (sheet "MutS-Tn first passage variants", see Methods).** Variants were called against the NCBI PAO1 reference (RefSeq accession NC_002516.2) and annotated with snpEff. Annotation fields are as follows: POS = Nucleotide coordinate position in PAO1 reference; REF = reference allele; ALT = alternate allele; EFFECT = variant classification (intergenic, missense, frameshift or synonymous); GENE = gene name when available, LOCUS_TAG = feature or neighboring features locus tag; DNA_MUT = nucleotide variant annotation; AA_MUT = amino acid change annotation when applicable.
(XLSX)

**S2 Data. An excel file with the detailed underlying data to generate the main and supplementary figures: Figs 1A, 1B, 2, 3A–3C, 4–4C, S1, S2, S3, S4, S5, S6, S7, S8, S9, and S11.**
(XLSX)

**S3 Data. General metrics for RNA-seq samples and libraries used in this study.** Ribosomal bases (%) and Median 5′ to 3′ bias were obtained with Picard Tools. RIN = RNA Integrity Number; GT = Genotype.
(XLSX)

**S4 Data. CZA concentration inhibiting population growth during in vitro adaptation and representative isolate E-test MICs.** Lineage and passage are encoded in isolate name in the following way: The first position indicates underlying genotype for each selection experiment (1 = MPAO1-WT; 2 = MPAO1-*mutS^{Tn}*; 3 = PT) followed by the lineage (A–D) and passage

number (1–20 for MPAO1-WT and 1–7 for MPAO1-*mutS*$^{Tn}$ and PT strains).
(XLSX)

**S5 Data. Variant annotation in isolates serially passaged on media without antibiotic.** Variants were called against the NCBI PAO1 reference (RefSeq accession NC_002516.2) and annotated with snpEff. Annotation fields are as follows: LINEAGE = underlying genotype in first position (1 = MPAO1-WT; 2 = MPAO1-*mutS*$^{Tn}$) followed by the lineage (A–D); POSITION = Nucleotide position; REFERENCE = reference allele; ALTERNATE = alternate allele; EFFECT = variant classification (intergenic, missense, frameshift or synonymous); FEATURE = coding or intergenic variant; FEATUREID = locus tag of affected CDS or neighboring CDS when intergenic; IMPACT = variant impact as predicted by snpEff; GENE NAME = gene name when available; BASE CHANGE = nucleotide change; AMINO ACID CHANGE = amino acid change.
(XLSX)

**S6 Data. List of genomes (strain and accession number) used in *Pseudomonas* gene ortholog analysis.**
(XLSX)

## Acknowledgments

We thank the staff of the Microbiology Service in the Dept. Lab Medicine, NIH Clinical Center for technical support and acknowledge the NISC Comparative Sequencing Program for providing high-throughput sequencing. We acknowledge Dr. Colin Manoil's laboratory for the construction of the transposon mutants used in this work (supported by grant # NIH P30 DK089507). The allelic exchange vector pEX18Gm and *E. coli* S17.1 (λ pir+) was a generous gift from Joe J Harrison, Department of Biological Sciences, University of Calgary, Calgary, Alberta, Canada. This work utilized the computational resources of the NIH HPC Biowulf cluster (http://hpc.nih.gov).

The content and views expressed in this work are those of the authors and do not necessarily represent the official views of the NIH, Department of Veterans Affairs, or US Government.

## Author Contributions

**Conceptualization:** Augusto Dulanto Chiang, Prashant P. Patil, Lidia Beka, Robert A. Bonomo, Pavel P. Khil, John P. Dekker.

**Data curation:** Augusto Dulanto Chiang, Prashant P. Patil, Lidia Beka, Jung-Ho Youn, Pavel P. Khil, John P. Dekker.

**Formal analysis:** Augusto Dulanto Chiang, Prashant P. Patil, Lidia Beka, Pavel P. Khil, John P. Dekker.

**Funding acquisition:** John P. Dekker.

**Investigation:** Augusto Dulanto Chiang, Prashant P. Patil, Lidia Beka, Jung-Ho Youn, Adrien Launay, Pavel P. Khil, John P. Dekker.

**Methodology:** Augusto Dulanto Chiang, Prashant P. Patil, Lidia Beka, Jung-Ho Youn, Adrien Launay, Pavel P. Khil, John P. Dekker.

**Project administration:** John P. Dekker.

**Resources:** John P. Dekker.

**Software:** Augusto Dulanto Chiang, Adrien Launay, Pavel P. Khil.

**Supervision:** Pavel P. Khil, John P. Dekker.

**Validation:** Augusto Dulanto Chiang, Prashant P. Patil, Lidia Beka, Pavel P. Khil, John P. Dekker.

**Visualization:** Augusto Dulanto Chiang, Prashant P. Patil, Lidia Beka, Pavel P. Khil, John P. Dekker.

**Writing – original draft:** Augusto Dulanto Chiang, Prashant P. Patil, Lidia Beka, Pavel P. Khil, John P. Dekker.

**Writing – review & editing:** Augusto Dulanto Chiang, Prashant P. Patil, Lidia Beka, Adrien Launay, Robert A. Bonomo, Pavel P. Khil, John P. Dekker.

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
