## [Editor Report · Decision Letter 0]

3 May 2022

Dear Dr Dekker, 

Thank you for submitting your manuscript entitled "Novel Mechanisms of Resistance to Ceftazidime/Avibactam and Ceftolozane/Tazobactam in Mismatch Repair-Deficient Pseudomonas aeruginosa" for consideration as a Research Article by PLOS Biology.

Your manuscript has now been evaluated by the PLOS Biology editorial staff, as well as by an academic editor with relevant expertise, and I am writing to let you know that we would like to send your submission out for external peer review.

Once your full submission is complete, your paper will undergo a series of checks in preparation for peer review. Once your manuscript has passed the checks it will be sent out for review. To provide the metadata for your submission, please Login to Editorial Manager (https://www.editorialmanager.com/pbiology) within two working days, i.e. by May 05 2022 11:59PM.

If your manuscript has been previously reviewed at another journal, PLOS Biology is willing to work with those reviews in order to avoid re-starting the process. Submission of the previous reviews is entirely optional and our ability to use them effectively will depend on the willingness of the previous journal to confirm the content of the reports and share the reviewer identities. Please note that we reserve the right to invite additional reviewers if we consider that additional/independent reviewers are needed, although we aim to avoid this as far as possible. In our experience, working with previous reviews does save time. 

If you would like to send previous reviewer reports to us, please email me at dummarino@plos.org to let me know, including the name of the previous journal and the manuscript ID the study was given, as well as attaching a point-by-point response to reviewers that details how you have or plan to address the reviewers' concerns. 

Kind regards,

Dario

Dario Ummarino, PhD

Senior Editor

PLOS Biology

dummarino@plos.org

---

## [Decision Letter · Decision Letter 1]

16 Jun 2022

Dear Dr. Dekker,

Thank you for your patience while your manuscript "Novel Mechanisms of Resistance to Ceftazidime/Avibactam and Ceftolozane/Tazobactam in Mismatch Repair-Deficient Pseudomonas aeruginosa" was peer-reviewed at PLOS Biology. It has now been evaluated by the PLOS Biology editors, an Academic Editor with relevant expertise, and by several independent reviewers. 

In light of the reviews, which you will find at the end of this email, we would like to invite you to revise the work to thoroughly address the reviewers' reports.

As you will see below, the reviewers find your work interesting, important and well executed, and appreciate that the paper is clearly written. However, they also raised several concerns that should be addressed before further consideration at PLOS Biology. In particular, we think it is important to address the concern of reviewer #2 that the main results may be anecdotal and hard to generalize. We consider that you should address all the reviewer comments, especially those regarding effects of mutation rate and population size. 

Given the extent of revision needed, we cannot make a decision about publication until we have seen the revised manuscript and your response to the reviewers' comments. Your revised manuscript is likely to be sent for further evaluation by all or a subset of the reviewers.

**IMPORTANT - SUBMITTING YOUR REVISION**

*Re-submission Checklist*

*Published Peer Review*

*PLOS Data Policy*

*Blot and Gel Data Policy*

Sincerely,

Paula

---

Senior Editor

PLOS Biology

REVIEWS:

Reviewer #1: Mechanisms of antibiotic resistance. 

Reviewer #2: Pathogen genomics and antibiotic resistance. 

Reviewer #3: Pseudomonas.

Reviewer #1: In this manuscript, Chiang et al. study the mechanisms of resistance that develops to CZA in Pseudomonas that have hypermutation due to mismatch repair deficiencies. CZA is an important expanded spectrum antibiotic that has the beta-lactamase avibactam which inhibits AmpC, an important resistance mechanism in Pseudomonas. The authors demonstrate that MMR-deficient Pseudomonas evolves resistance through mechanisms that are not seen in WT lineages. This includes the finding that mutations in MexVW increase resistance to CZA which was found in evolved strains and then with mutations engineered into a WT background. This alternative resistance mechanism is certainly underappreciated and is an important finding. The paper is clear, thorough and well written. A few points to be improved/clarified upon include:

1. In the search for mutations in MexV and MexW it was noted that the mutations in this study were not found in sequenced isolates in public databases. Can the authors discuss the number of isolates in the database that had mutS mutations present? It would be helpful to know what subset of available genomes are thought to be deficient in MMR.

2. In addition, the authors note an early inactivating MexB mutation W735R. It would likewise be interesting to know how common the mutation is seen. If present in other genomes, how common was it?

3. The authors state in the discussion "Given these findings, identification of hypermutator phenotypes in the clinical microbiology lab, either through direct phenotypic testing or through targeted gene amplification and sequencing, may in the future inform optimized treatment decisions based on these differences in behavior." This is an important point. However, I think it needs to be put in the context of the percentage of time that hypermutators are differentially resistant to CZA. Did the authors happen to look back at the public data (in addition to the question in #1 above) and cross reference with any available AST data for CZA in those sequenced strains to determine how often CZA resistance is found (there might not be much out there)? This will be critical to making the point that knowing mutS status for example could translate into different clinical decisions. 

4. What if anything is known about the existing transposon mutants for MexV and MexW in the same collection that was used for this study? Were they tested to see if the MIC to CZA changes and are the transposon insertions covering the same mutations that were found in the evolution experiments? The real point of this question is to get at whether the mutations that were found in MexV and MexW are the critical bases to determine a phenotype or whether a variety of mutations in these genes could lead to a similar phenotype.

Reviewer #2: The manuscript explores mutational paths to betalactam (ceftazidime/avibactam) resistance in a hypermutating MMR-deficient Pseudomonas aeruginosa strain and compares them to paths taken by non-mutator strains. They find that the hypermutating strain can attain much higher resistance than the wild-type and does so largely through unexpected mechanisms. Specifically, it seems that an initial random inactivating mutation of the most relevant efflux pump (MexAB) drove the hypermutating populations to alternative resistance paths, where the resistance can at least partly be attributed to an upregulation of an alternative, and less-known, MexVW efflux pump. The authors then confirm that the resistance paths through MexAB explain the resistance evolved in the wild-type and that MexVW gives a comparable advantage through an independent mechanism. The mansucript is written clearly and meaningfully combines experimental evolution, sequencing, transcriptional profiling and comparisons to deposited sequences to understand the evolutionary potential of Pseudomonas aeruginosa hypermutators. However, some of the claims need more quantitative statistical support. Also, it is not clear how well these findings can generalize to other situations, since the most important finding comes down to a mutation which may have been anecdotal.

General comments:

- There is some confusion as to how much of the extremely successful adaptation of the mutator is due to the fact that the mutation rate is higher and therefore the populations are not limited in time by waiting for the next mutation occurring, and how much is thanks to the fact that non-beneficial mutations can easily fix in mutators, opening doors to very different, and potentially more successful, adaptation paths. It seems the interpretations go exclusively in the direction of the second, which is more interesting, but I think some of the results could be explained with only the first. To be more specific, would the mexVW mutations evolve in also the wildtype if given enough time, since they carry a benefit also without the deactivating mexB mutation (Supp. Fig. 9)? And if the mexB mutation is indeed important, is it sufficient? If any mexB loss of function mutation occurred on a non-mutator background, would it stimulate such quick adaptation by itself? Even speculations that would answer these questions from the authors would be appreciated, if experiments seem outside of the scope.

- Another general worry is that the reported findings, although very interesting, seem anecdotal and difficult to generalize. Is there a reason to think that other truly independent starting cultures of this or a different mutator would also get the deactivating mexB mutation with any notable frequency? Would they then go on this same new mexVW resistance path? Are there any sequenced hypermutating clinical isolates that would confirm the relevance of this resistance path and its link to a mutator background?

- Some relatively important claims are not backed up by a figure and quantitative analysis:

 - - Figure 2 does not highlight the data that supports the main statement: mutators have mutations in genes unrelated to common antibiotic resistance mechanisms (e.g. betalactamases or efflux pumps). Instead of (or in addition to) the conservation and amino-acid similarity scores, the mutations should be labeled by whether they have or have not occurred in other published evolution experiments, and/or whether these genes are associated with known resistance mechanisms (e.g. differentiating whether they are expected to confer resistance also for this particular antibiotic or to others). 

 - - Supplementary table 4 - this data needs to be shown in a figure. The claim seems to be that the mutations in the lineages evolved in antibiotic were different to the ones without, but this is not clearly supported. It also needs to be clear how many separate lineages were passaged.

Minor comments: 

- Not clear what was expected for figure 1b vs what was found. Either a clearer explanation of what to look for in this plot or a different general analysis, e.g. of gene ontology/function would be more informative.

- The information about the number of separate lineages of each background that were evolved is not clear enough. Especially Fig 1 calls for this type of information.

- Line 170. The important fraction here is not the number of mutations associated with resistance compared to all mutations that occurred in mutS. A much higher mutation rate will for sure yield many "hitchhiking" mutations which are not expected to have any benefit. The number of "relevant" targets in the mutator strain could potentially be compared to the number of fixed mutations in the wild type that underwent a similar increase in fold-resistance.

- Line 196-197. I think there is a similar flaw in interpretation here. Shared mutations in mutator lineages do not suggest selective advantage if there is any chance of a common source of the mutation (which there clearly is here). High frequency of synonymous mutations, as shown in e.g. Fig 1a, support the fact that rate of hitchhiking was high and many fixed mutations are not expected to be beneficial. Only truly independently (ideally different loci) acquired mutations in the same gene targets could provide this type of support.

- Line 292-293 I don't see why it would preclude direct comparison, the satellite colonies are expected with highly mutating strains, the zone where the lawn-forming bacteria are inhibited is the one to read out, after the recommended 16h this looks plausible to me. It would be very informative to add the MIC values of the evolved mutators to the plot in Supp. Figure 9.

Reviewer #3: Comments to the authors

The manuscript by Dulanto et al. unravels the diverse evolutionary pathways that bacterial mutator clones can explore for the acquisition of antibiotic resistance in the opportunistic pathogen Pseudomonas aeruginosa. The authors perform in vitro adaptive evolution experiments by challenging reference PAO1 strain and a mutS isogenic mutant with increasing concentrations of ceftazidime-avibactam (CAZ). The experiments confirm that acquisition of CZA resistance is boosted by hypermutation compared with the wild-type non mutator strain. By applying whole-genome sequencing and transcriptomic of initial and final bacterial clones of the several mutator and wild-type lineages, the authors reveal a novel mechanism of mutation-driven β-lactam resistance based on the multidrug efflux pump MexVW, which has been previously described to confer resistance to antibiotics other than β-lactams. In contrast to wild-type lineages that are selected for mutations altering well-known CAZ resistance genes, including those leading to MexAB efflux pump overexpression, the work describes that MexVW-mediated resistance was acquired only by mutator lineages. Surprisingly, the authors find that the mutator inoculum hold an inactivating mutation in the mexB gene. Based upon this observation, they wonder if the MexVW resistance pathway evolved in mutators as an alternative mechanism, facing the primary mexB mutation that prevents the acquisition of resistance by MexAB efflux pump.

The contribution of this manuscript in P. aeruginosa resistance evolution processes associated with hypermutability is significant. The data provided support the conclusions. The manuscript is clear in communicating the main findings of the study.

Major points

A major concern is respect to the size of the initial inoculum used for the experimental evolution assays. The smaller the initial inoculum, the smaller the probability to contain any preexisting mutant. This is particularly important when handling mutator strains and de novo mutations aim to be analyzed under the experimental conditions. To overcome this issue serial dilutions should be performed in order to obtain inocula ranging from 10 to 1000 cells. Here, authors have used a dilution of 0.05 OD600 (Line 502), which is a quite large inoculum that explains the heterogeneity of the initial population the authors observed.

Still, in this case, one of these preexisting mutations in mutator lineages inoculum resulted to be mexB. As authors hypothesize, mexB background may lead to mutator lineages bias their evolution towards alternative resistance genetic pathway: the central message of the work. Although authors support this hypothesis determining that mexB preexisting mutation fully inactivates the main MexAB-mediated resistance mechanism, what would happen if evolution experiments were conducted with small mutator inoculums avoiding mexB mutations in the initial populations? Likewise, what would happen if the experimental evolution were conducted from a normomutator mexB mutant? These approaches could help further supporting the hypothesis

Minor points

1) At least in the first paragraphs of the Introduction section that I could check in detail, it seems that many references do not specifically correspond with the stated text. For instance:

-Lines 60-62: This sentence refers to hypermutation in chronic infection with P. aeruginosa that is underlain by the inactivation of genes involved in DNA repair.

Taddei 1997 and Yang 2008, could be replaced by some more accurate such as Oliver et al. (2000, Science 288: 1251-1254); Mandsberg et al. (2009, AAC 53: 2483-2491); Ciofu et al. (2005, AAC 49: 2276-2282).

I suggest the references be checked.

2) Data and references related to the clinical PT strain are missing in Table S8.

3) Numbers in Line 113 do not match with those in Figure 1a.

4) Is PDC deletion described in Table S3 (referred from line 124) the same that those described in lines 128-130). Anyway, details of PDC deletion in Table S3 should be included as well as the other indel included in the Table.

5) Table S3 only includes data from PT strain lineages. It could be quite informative the detail of all mutations occurred in genes related to β-lactam resistance in all analyzed lineages (MPAO1 wt, MPAO1 mutS and PT)

---

## [Decision Letter · Decision Letter 2]

22 Sep 2022

Dear Dr. Dekker,

Thank you for your patience while we considered your revised manuscript "Novel Mechanisms of Resistance to Ceftazidime/Avibactam and Ceftolozane/Tazobactam in Mismatch Repair-Deficient Pseudomonas aeruginosa" for publication as a Research Article at PLOS Biology. This revised version of your manuscript has been evaluated by the PLOS Biology editors, the Academic Editor, and the original reviewers.

Based on the reviews, we are likely to accept this manuscript for publication, provided you satisfactorily address the remaining points raised by the reviewers. In particular, reviewer #2 wants you to add some more argumentation about why the mutations found in the hypermutating strain are beneficial. This reviewer also suggests to include in the manuscript your response to the comment about mexB deactivation frequency. Please also make sure to address the following data and other policy-related requests.

1. We recommend a change in the title: "Hypermutator strains of Pseudomonas aeruginosa reveal novel pathways of resistance to combinations of cephalosporin antibiotics and beta-lactamase inhibitors ". This is a suggestion, so please modify as you think it fits better.

2. DATA POLICY:

You may be aware of the PLOS Data Policy, which **requires that all data be made available without restriction**: http://journals.plos.org/plosbiology/s/data-availability. For more information, please also see this editorial: http://dx.doi.org/10.1371/journal.pbio.1001797

A) Supplementary files (e.g., excel). Please ensure that all data files are uploaded as 'Supporting Information' and are invariably referred to (in the manuscript, figure legends, and the Description field when uploading your files) using the following format verbatim: S1 Data, S2 Data, etc. Multiple panels of a single or even several figures can be included as multiple sheets in one excel file that is saved using exactly the following convention: S1_Data.xlsx (using an underscore).

B) Deposition in a publicly available repository. Please also provide the accession code or a reviewer link so that we may view your data before publication.

Regardless of the method selected, please ensure that you provide the individual numerical values that underlie the summary data displayed in the following figure panels as they are essential for readers to assess your analysis and to reproduce it: Figures 1AB, 2, 3ABC, 4ABC, and Supplementary Figures SF1, SF2, SF3, SF4, SF5, SF6, SF7, SF8, SF9, SF11.

**Please also ensure that figure legends in your manuscript include information on where the underlying data can be found, and ensure your supplemental data file/s has a legend.**

We expect to receive your revised manuscript within two weeks.

*Published Peer Review History*

*Press*

Sincerely,

Paula

---

Senior Editor,

pjaureguionieva@plos.org,

PLOS Biology

Reviewer remarks:

Reviewer #1: Antibiotic resistance in P. aeruginosa.

Reviewer #2: Evolution of antibiotic resistance.

Reviewer #3: Adaptive mutagenic mechanisms in P. aeruginosa.

Reviewer #1: The authors have responded to my comments and questions in a thorough manner. The additional analysis looking for the presence of some of these mutations in the public database is an additional strength to the manuscript.

Reviewer #2: I appreciate the thought and work put into addressing my comments and indeed, all major points have been addressed. The most important was the literature search and analysis confirming that mutators, mexB deactivating mutations and likely even their combinations are common in clinical strains. This does make it plausible that the scenario the authors find in the evolution experiments would be relevant for the clinic as well. Only very few minor points remain:

-In the now revised explanation of why the authors believe the mutations found in the hypermutating strain are indeed beneficial (starting line 211 in tracked version), there is still a flaw. Even the increase in frequency of a mutation under antibiotic selection cannot by itself suggest each such mutation's role in antibiotic resistance. It is clear that the mutator acquired some mutations that gave it resistance, but very likely the remaining ones (likely a majority) just happened to appear on the background of beneficial ones and that is the reason for their increase in frequency, not their effect on fitness. Specifically in hypermutating strains, this is extremely common. This is mainly a point to improve the argumentation. In principle, I agree with the authors that it is very likely that some of the mutations identified in the mutator strains are actually previously unidentified resistance mutations. It is very difficult to understand which ones, just from the experiments conducted, so this warrants further study, outside the scope of this manuscript.

-In the response to my comment about mexB deactivation frequency, the authors argue that mexB mutations are likely to be wrongly classified as mexB activating/modifying and not deactivating as they confirm. I could not find this note in the updated version of the manuscript (somehow the line numbers did not match in the tracked file, so I am not 100% sure), but think is very interesting, so would be worth including if the space limitations allow.

Reviewer #3: The authors have improved the manuscript accordingly with the comments and suggestions expressed by the reviewers. I'm pleased with the changes included in the revised version of the article as well as with all new data, analyses and arguments they detaily addressed in the response letter.

---

## [Editor Report · Decision Letter 3]

13 Oct 2022

Dear Dr. Dekker,

Thank you for the submission of your revised Research Article "Hypermutator strains of Pseudomonas aeruginosa reveal novel pathways of resistance to combinations of cephalosporin antibiotics and beta-lactamase inhibitors" for publication in PLOS Biology. On behalf of my colleagues and the Academic Editor, Tobias Bollenbach, I am pleased to say that we can in principle accept your manuscript for publication, provided you address any remaining formatting and reporting issues. These will be detailed in an email you should receive within 2-3 business days from our colleagues in the journal operations team; no action is required from you until then. Please note that we will not be able to formally accept your manuscript and schedule it for publication until you have completed any requested changes.

PRESS

Sincerely, 

Paula

---

Senior Editor

PLOS Biology
